# Algorithms for Caching and MTS with reduced number of predictions*

**Karim Abdel Sadek**
University of Amsterdam†
karim.abdel.sadek@student.uva.nl

**Marek Eliáš**
Department of Computing Sciences
Bocconi University
marek.elias@unibocconi.it

## Abstract

ML-augmented algorithms utilize predictions to achieve performance beyond their worst-case bounds. Producing these predictions might be a costly operation – this motivated Im et al. (2022) to introduce the study of algorithms which use predictions parsimoniously. We design parsimonious algorithms for caching and MTS with *action predictions*, proposed by Antoniadis et al. (2023), focusing on the parameters of consistency (performance with perfect predictions) and smoothness (dependence of their performance on the prediction error). Our algorithm for caching is 1-consistent, robust, and its smoothness deteriorates with the decreasing number of available predictions. We propose an algorithm for general MTS whose consistency and smoothness both scale linearly with the decreasing number of predictions. Without the restriction on the number of available predictions, both algorithms match the earlier guarantees achieved by Antoniadis et al. (2023).

## 1 Introduction

Caching, introduced by Sleator and Tarjan (1985), is a fundamental problem in online computation important both in theory and practice. Here, we have a fast memory (cache) which can contain up to $k$ different pages and we receive a sequence of requests to pages in an online manner. Whenever a page is requested, it needs to be loaded in the cache. Therefore, if the requested page is already in the cache, it can be accessed at no cost. Otherwise, we suffer a *page fault*: we have to evict one page from the cache and load the requested page in its place. The page to evict is to be chosen without knowledge of the future requests and our target is to minimize the total number of page faults.

Caching is a special case of Metrical Task Systems introduced by Borodin et al. (1992) as a generalization of many fundamental online problems. In the beginning, we are given a metric space $M$ of states which can be interpreted as actions or configurations of some system. We start at a predefined state $x_0 \in M$. At time steps $t = 1, 2, \ldots$, we receive a cost function $\ell_t \colon M \to \mathbb{R}^+ \cup \{0, +\infty\}$ and we need to make a decision: either to stay at $x_{t-1}$ and pay a cost $\ell_t(x_{t-1})$, or to move to another, possibly cheaper state $x_t$ and pay $\ell_t(x_t) + d(x_{t-1}, x_t)$, where the distance $d(x_{t-1}, x_t)$ represents the transition cost between states $x_{t-1}$ and $x_t$.

The online nature of both caching and MTS forces an algorithm to make decisions without knowledge of the future which leads to very suboptimal results in the worst case (Borodin et al., 1992; Sleator and Tarjan, 1985). A recently emerging field of learning-augmented algorithms, introduced in seminal papers by Kraska et al. (2018) and Lykouris and Vassilvitskii (2021), investigates approaches to improve the performance of algorithms using predictions, possibly generated by some ML model. In general, no guarantee on the accuracy of these predictions is assumed. Therefore, the performance of learning-augmented algorithms is usually evaluated using the following three parameters:

*Consistency.* Performance with perfect predictions, preferably close to optimum.

*Robustness.* Performance with very bad predictions, preferably no worse than what is achievable by known algorithms which do not utilize predictions.

---

*Full version of this paper can be found in Appendix and at https://arxiv.org/abs/2404.06280

†The presentation of this paper was financially supported by the Amsterdam ELLIS Unit and Qualcomm. Work completed while Abdel Sadek was in his final year of BSc at Bocconi University

*Smoothness.* Algorithm's performance should deteriorate smoothly with increasing prediction error between the consistency and robustness bound.

These three parameters express a desire to design algorithms that work very well when receiving reasonably accurate predictions most of the time and, in the rest of the cases, still satisfy state-of-the-art worst-case guarantees. See the survey by Mitzenmacher and Vassilvitskii (2020) for more information.

Producing predictions is often a computationally intensive task, therefore it is interesting to understand the interplay between the number of available predictions and the achievable performance. In their inspiring work, Im et al. (2022) initiated the study of learning-augmented algorithms which use the predictions parsimoniously. In their work, they study caching with next-arrival-time predictions introduced by Lykouris and Vassilvitskii (2021). Their algorithm uses $O(b \log_{b+1} k)$ OPT predictions, where OPT is the number of page faults incurred by the offline optimal solution and $b \in \mathbb{N}$ is a parameter. It achieves smoothness linear in the prediction error. It satisfies tight consistency bounds: with perfect predictions, it incurs at most $O(\log_{b+1} k)$ OPT page faults and no algorithm can do better. In other words, it achieves a constant competitive ratio with unrestricted access to predictions ($b = k$) and, with $b$ a small constant, its competitive ratio deteriorates to $O(\log k)$ which is comparable to the best competitive ratio achievable without predictions. One of their open questions is whether a similar result could be proved for MTS.

In this paper, we study parsimonious algorithms for MTS working with *action predictions* which were introduced by Antoniadis et al. (2023). Here, each prediction describes the state of an optimal algorithm at the given time step and its error is defined as the distance from the actual state of the optimal algorithm. The total prediction error is the sum of errors of the individual predictions. In the case of caching, action predictions have a very concise representation, see Section 2.1. Unlike next-arrival-time predictions, action predictions can be used for any MTS. Using the method of Blum and Burch (2000), it is easy to achieve near-optimal robustness for any MTS losing only a factor $(1 + \epsilon)$ in consistency and smoothness. Therefore, we study how the reduced number of predictions affects the consistency and smoothness parameters. We consider the following two regimes.

*Bounded number of predictions:* The algorithm can request a prediction whenever it prefers as far as the total number of requested predictions is bounded by $b$ OPT, where $b$ is a parameter. This regime is similar to Im et al. (2022).

*Well-separated queries to the predictor:* The queries to the predictor need to be separated by at least $a$ time steps, for some parameter $a$. This captures the situation when producing each prediction takes more than one time step.

## 1.1 OUR RESULTS

We evaluate the algorithm's performance using *competitive ratio* which is, roughly speaking, the worst-case ratio between the cost incurred by the algorithm and the cost of the offline optimum, see Section 2 for a formal definition. We say that an algorithm achieves consistency $\alpha$ and robustness $\beta$ if its competitive ratio is at most $\alpha$ when provided with perfect predictions and at most $\beta$ with arbitrarily bad predictions. For a given function $g$, we call an algorithm $g(\eta)$-smooth if its competitive ratio is at most $g(\eta)$ whenever provided with predictions with the total error at most $\eta$.

Our first contribution is an algorithm for caching which receives action predictions describing the states of the optimal offline algorithm Belady proposed by Belady (1966). High quality such predictor based on imitation learning was already designed by Liu et al. (2020). Its empirical evaluation within existing algorithms designed for action predictions was performed by Chledowski et al. (2021).

**Theorem 1.1.** *Let $f$ be an increasing convex function such that $f(0) = 0$ and $f(i) \leq 2^i - 1$ for each $i \geq 0$. There is an algorithm for caching requiring $O(f(\log k))$ OPT predictions which achieves consistency 1, robustness $O(\log k)$, and smoothness $O(f^{-1}(\eta/OPT))$, where $\eta$ denotes the total prediction error with respect to Belady and OPT is the number of page faults of Belady.*

In fact, the number of required predictions is slightly smaller than what is stated in the theorem. Table 1 shows numbers of predictions and achieved smoothness for some natural choices of $f$. Already with $O(\sqrt{k})$ OPT predictions, our bounds are comparable to Antoniadis et al. (2023) whose algorithm asks for a prediction in every step, its consistency is constant and its smoothness is logarithmic in

Table 1: Smoothness vs. number of predictions.

| $f(i)$ | $2^i - 1$ | $i^2$ | $i$ | $0$ |
|---|---|---|---|---|
| # of predictions | $O(\sqrt{k})\,\mathrm{OPT}$ | $O(\log^2 k)\,\mathrm{OPT}$ | $O(\log k)\,\mathrm{OPT}$ | $2\,\mathrm{OPT}$ |
| smoothness | $O(1 + \log(\frac{\eta}{\mathrm{OPT}} + 1))$ | $O(\sqrt{2\frac{\eta}{\mathrm{OPT}}})$ | $O(\frac{\eta}{\mathrm{OPT}})$ | $O(\frac{k\eta}{\mathrm{OPT}})$ |

$\eta$. The algorithm also works with $f(i) = 0$. In that case, it asks for at most $2\,\mathrm{OPT}$ predictions and still remains 1-consistent. However, its smoothness is not very good. We use sliding marking phases and a careful distribution of queries of the predictor over the time horizon. This allows us to avoid dealing with so called "ancient" pages considered by Rohatgi (2020) and Antoniadis et al. (2023), resulting in an algorithm with better consistency and a simpler analysis.

We discuss tightness of our bounds in Section 7 in the full version of this paper (see Appendix). We show that with, for example, only $0.5 OPT$ available predictions, no algorithm can be better than $O(\log k)$-competitive – a guarantee comparable to the best classical online algorithms without predictions. We also show that the number of predictions used by our algorithm is close to optimal.

**Theorem 1.2.** *Let $f$ be an increasing function. Any $f(\eta)$-smooth algorithm for caching with action predictions, i.e., an algorithm whose competitive ratio with predictions of error $\eta$ is $f^{-1}(\eta)$ for any $\eta > 0$, has to use at least $f(\ln k)\,\mathrm{OPT}$ predictions.*

For general MTS, we cannot bound the number of used predictions as a function of OPT. The reason is that any instance of MTS can be scaled to make OPT arbitrarily small, allowing us to use only very few predictions. We propose an algorithm which queries the predictor once in every $a$ time steps, making at most $T/a$ queries in total, where $T$ denotes the length of the input sequence.

**Theorem 1.3.** *There is a deterministic algorithm for any MTS which receives a prediction only once per each $a$ time steps and its cost is at most $O(a) \cdot (\mathrm{OFF} + 2\eta)$, where $\mathrm{OFF}$ denotes the cost of an arbitrary offline algorithm and $\eta$ the error of predictions with respect to this algorithm.*

This is a more general statement than Theorem 1.1 which requires OFF to be Belady. Considering any offline optimal algorithm OFF, Theorem 1.3 implies a smoothness $O(a) \cdot (1 + 2\eta/\mathrm{OPT})$ and consistency $O(a)$. Our algorithm is based on work functions. For $a = 1$, its smoothness is $1 + 2\eta/\mathrm{OFF}$, see Section 4, which improves upon the smoothness bound of $1 + 4\eta/\mathrm{OFF}$ by Antoniadis et al. (2023). It is not robust on its own. However, it can be combined with any online algorithm for the given MTS using the result of Blum and Burch (2000) achieving robustness comparable to that algorithm and losing only a factor of $(1 + \epsilon)$ in smoothness and consistency.

No algorithm receiving a prediction only once in $a$ time steps can be $o(a)$-consistent. This follows from the work of Emek et al. (2009) on advice complexity, see Section 7 of the full version (in Appendix) for more details. The same can be shown for smoothness by modifying the lower bound construction of Antoniadis et al. (2023).

**Theorem 1.4.** *There is no $o(a\eta/\mathrm{OPT})$-smooth algorithm for MTS with action predictions which receives predictions only once in $a$ time steps.*

We can modify our algorithm for caching to ensure that the moments when the predictions are queried are separated by at least $a$ time steps, not losing too much of its performance.

**Theorem 1.5.** *There is an algorithm for caching which receives prediction at most once in $a \leq k$ time steps and using at most $O(f(\log k))\,\mathrm{OPT}$ predictions in total which is $O(1)$-consistent, $O(\log k)$-robust and $O(f^{-1}(a\eta/\mathrm{OPT}))$-smooth.*

In Section 5, we provide empirical results suggesting that our algorithm's performance can be comparable to the performance of algorithms imposing no limitations on their use of predictions. Our algorithm may therefore be useful especially with heavy-weight predictors like (Liu et al., 2020).

In Section 8 of the full version of this paper (see Appendix), we provide an algorithm for an alternative prediction setup which we call FitF oracle: each prediction says which of the pages in the current algorithms cache will be requested furthest in the future.

## 1.2 RELATED WORK

The most related work is by Im et al. (2022), who studied caching with next arrival time predictions. A smaller number of predictions affects the consistency of their algorithm: with $b(\log k / \log b)$ OPT predictions, they achieve consistency $O(\log k / \log b)$ and they show that this is tight. They also show that their algorithm achieves linear smoothness. In contrast, our algorithm is 1-consistent when receiving at least OPT predictions. This demonstrates that action predictions, although not containing more bits, seem to contain useful information about the input instance in a more condensed form. See (Antoniadis et al., 2023) for comparison and connections between these prediction setups. Drygala et al. (2023) study ski rental and bahncard problems with predictions of a fixed cost.

There are several other papers on caching with predictions, including (Lykouris and Vassilvitskii, 2021; Rohatgi, 2020; Wei, 2020; Emek et al., 2009; Antoniadis et al., 2023; 2022) which design algorithms asking for a prediction at each time step. Consistency parameters achieved by these algorithms are constants greater than 1. Note that those using black-box methods to achieve robustness are $(1 + \epsilon)$-consistent (e.g. (Wei, 2020)). We can explicitly compare our smoothness bounds to Antoniadis et al. (2023) who use the same kind of predictions: their smoothness is $O(1 + \log(\frac{\eta}{\text{OPT}} + 1))$ with unlimited use of predictions while our algorithm achieves the same smoothness bound with $O(\sqrt{k})$ OPT predictions. We compare the smoothness of the other algorithms experimentally in Section 5. Antoniadis et al. (2022) study a prediction setup where each prediction is only a single bit, however their algorithms need to receive it in every time step. Gupta et al. (2022) study several problems including caching in a setting where each prediction is correct with a constant probability.

Antoniadis et al. (2023) proposed a 1-consistent and $(1 + 4\eta/\text{OPT})$-smooth algorithm for MTS with action predictions which can be robustified by loosing factor $(1 + \epsilon)$ in consistency and smoothness. Getting smoothness bounds sublinear in $\eta/\text{OPT}$ for specific MTS problems other than caching remains a challenging open problem even with unlimited number of predictions and this holds even for weighted caching. Specific results on weighted caching are by Jiang et al. (2022) who studied it in a setup requiring very verbose predictions and by Bansal et al. (2022) whose bounds depend on the number of weight classes. There is also a consistency/robustness trade-off by Lindermayr et al. (2022) for $k$-server.

Since the seminal papers by Kraska et al. (2018) and Lykouris and Vassilvitskii (2021) which initiated the study of learning-augmented algorithms, many computational problems were considered. There are papers on ski rental (Purohit et al., 2018), secretary problem (Dütting et al., 2021), online TSP (Bernardini et al., 2022), energy efficient scheduling (Bamas et al., 2020), flow-time scheduling (Azar et al., 2021; 2022), and online page migration (Indyk et al., 2022). Further related works can be found in References and are discussed in the full version of this paper (see Appendix).

## 2 PRELIMINARIES

Consider an algorithm ALG for MTS which produces a solution $x_0, x_1, \ldots, x_T$ for an instance $I$ consisting of cost functions $\ell_1, \ldots, \ell_T$. We denote $\text{cost}(\text{ALG}(I)) = \sum_{t=1}^{T} (\ell_t(x_t) + d(x_{t-1}, x_t))$. We say that ALG is $r$-competitive with respect to an offline algorithm OFF if there is an absolute constant $\alpha \in \mathbb{R}$ such that $\mathbb{E}[\text{cost}(\text{ALG}(I))] \leq r \cdot \text{cost}(\text{OFF}(I)) + \alpha$ for any instance $I$. If ALG is $r$-competitive with respect to an optimal offline algorithm, we say that ALG is $r$-competitive and call $r$ the competitive ratio of ALG. In the classical setting (without predictions), the best achievable competitive ratios are $\Theta(\log k)$ for caching (Fiat et al., 1991) and of order $poly \log n$ for MTS (Bartal et al., 2006; Bubeck et al., 2019), where $n$ is the number of points in the underlying metric space $M$. We refer to (Borodin and El-Yaniv, 1998) for a textbook treatment.

### 2.1 ACTION PREDICTIONS FOR MTS

Antoniadis et al. (2023) proposed a prediction setup which they call *action predictions*, where the predictions tell us what a good algorithm would do. More precisely, at each time $t$, the algorithm receives a prediction $p_t$ of a state where some offline algorithm OFF moves to at time $t$. The error of prediction $p_t$ is then $\eta_t = d(p_t, o_t)$, where $o_t$ is the real state of OFF at time $t$. The total prediction error is defined as $\eta = \sum_{t=1}^{T} \eta_t$.

Considering the case of caching, the state corresponds to a cache content, and the prediction error is the number of pages present in the cache of OFF and absent from the predicted cache content. A whole cache content may seem like a huge piece of information, but action predictions for caching can be implemented in a very succinct way. Antoniadis et al. (2023) explain how to represent them with only $O(\log k)$ bits per time step when they are received at each time step. Our algorithm asks, in each query, a specific number of indices of pages which are present in its cache but absent from the predicted cache. When we talk about a bound on the number of provided predictions, this bound applies both to the number of such queries as well as to the total number of indices reported by the predictor during the running time of the algorithm. There are predictors which can generate predictions of a similar kind by Jain and Lin (2016); Shi et al. (2019); Liu et al. (2020). See (Antoniadis et al., 2023) for a detailed treatment of this prediction setup and a comparison to other setups for caching.

## 2.2 CACHING: BELADY'S ALGORITHM, MARKING, AND LAZY ALGORITHMS

The classical optimal offline algorithm for caching proposed by Belady (1966) is denoted Belady in this paper. At each page fault, it evicts a page which is going to be requested furthest in the future. In the case of a tie, i.e., if there are several pages in the cache which will not be requested anymore, it chooses one of them arbitrarily. Our caching algorithm assumes that the predictor is trying to simulate Belady. The following useful property allows us to detect errors in the predictions quickly. It was recently used by Eliáš et al. (2024).

**Observation 2.1.** Consider request sequence $r_1, \ldots, r_T$. For any $t \leq T$, the cost incurred by Belady for $r_1, \ldots, r_T$ until time $t$ is the same as the cost of Belady with input $r_1, \ldots, r_t$.

To see this, it is enough to note that the solution produced by Belady with input $r_1, \ldots, r_T$ agrees until time $t$ with the solution produced by Belady on $r_1, \ldots, r_t$ which breaks ties based on the arrival times in $r_{t+1}, \ldots, r_T$.

We use properties of *marking* algorithms in this work. Such algorithms split the input sequence into phases, i.e., maximal subsequences where at most $k$ distinct pages are requested. Usually, the first phase starts in the beginning and the next phase follows just after the end of the previous one. However, we will consider phases starting at arbitrary moments. Let $O$ be the cache content of an algorithm in the beginning of the phase. Whenever a page is requested for the first time during the phase, we call this moment an *arrival* and we *mark* the page. At the end of the phase, the set $M$ of marked pages will have size $k$: some of them belong to $O$ and are called *old* while those in $C = M \setminus O$ are called *clean*. Exactly $|C|$ pages from $O$ remain unmarked until the end of the phase.

Marking algorithms is a class of algorithms which never evict a marked page and all of them have cache content $M$ at the end of the phase. Belady is not marking and our algorithm is not marking either, although it uses ideas from marking to achieve desired robustness and smoothness properties. At the end of each phase, we can bound the difference between the cache content of some algorithm and marking.

**Observation 2.2.** Let $c$ be the cost incurred by some algorithm during a marking phase. Then, $c \geq |M \setminus S|$, where $S$ is the cache content of the algorithm at the end of the phase and $M$ is the set of $k$ pages requested during the phase.

This is because each page in $p \in M$ has to be present in algorithm's cache when requested during the phase. If $p \notin S$, then the algorithm must have evicted it during the phase incurring cost 1.

**Observation 2.3.** If a page $p$ is evicted by Belady at time $t$, then $p$ is not going to be requested in the marking phase containing $t$ anymore.

If $p$ is evicted by Belady at time $t$, then the currently requested page $r_t$ and $k - 1$ pages from the cache are $k$ distinct pages that are requested before the moment when $p$ is requested next time. The current phase then needs to end before that moment.

We say that an algorithm is *lazy* if it evicts only one page at a time and only at a page fault. Belady is lazy while our algorithm, as described, may not be. However, any algorithm can be made lazy without increasing its cost. See (Borodin and El-Yaniv, 1998) for more information about caching.

**Observation 2.4.** The difference in the cache content of two lazy algorithms can increase only if both of them have a page fault. In that case, it can increase by at most 1.

## 3    BOUNDED NUMBER OF PREDICTIONS

In this section, we prove Theorem 1.1. We propose an algorithm called F&R which consists of two parts: Follower and Robust. It starts with Follower which is 1-consistent, but lacks in smoothness and robustness. At each page fault, Follower recomputes Belady for the part of the sequence seen so far and checks whether it also has a page fault. If yes, it copies the behavior of the predictor (Line 3). Otherwise, it must have received an incorrect prediction before. Therefore, it switches to Robust (Line 5) which is no more 1-consistent, but achieves desired smoothness and robustness. Robust runs for one marking phase and then returns back to Follower. At such moment, the predictor's and the algorithm's cache can be very different and Follower may need to lazily synchronize with the predictor (Line 4).

---

**Algorithm 1:** Follower

1  $P :=$ initial cache content;                                  // Prediction for time 0
2  **foreach** *pagefault* **do**
3      **if** $r_t \notin P$ *and* Belady *has a pagefault* **then**  query new prediction $P$ and evict any $p \in C \setminus P$;
4      **else if** $r_t \in P$ **then**  evict arbitrary $p \notin P$;
5      **else**  Switch to Robust (Algorithm 2);

---

Algorithm Robust runs during a single marking phase starting at the same moment, splitting it into windows as follows (assuming $k$ is a power of 2): The first window $W_1$ starts at the beginning of the phase and lasts $k/2$ arrivals, i.e., it ends just before the arrival number $k/2 + 1$. $W_i$ follows the $W_{i-1}$ and its length is half of the remaining arrivals in the phase. The last window $W_{\log k+1} = \{k\}$ lasts until the end of the phase. Robust comes with an increasing convex function $f$ such that $f(0) = 0$ and $f(i) \leq 2^j - 1$. Faster growing $f$ does not further improve our smoothness bounds. Function $f$ determines that we should request $f(i) - f(i-1)$ predictions in window $i$. If the window is too small, we ask for prediction at each time step. Robust starts with the cache content of a marking algorithm whose new phase would start at the same moment (Line 1). In the case of a page fault, it evicts an unmarked page chosen uniformly at random. At arrivals belonging to the sets $S$ and $F$, it performs synchronization with the predictor and queries the predictor's state respectively. The synchronization is always performed with respect to the most recent prediction $P$ which, in the case of lazy (or lazified) predictors, implicitly incorporates information from the previous predictions.

---

**Algorithm 2:** Robust$_f$ (one phase)

1  Load $k$ distinct most recently requested pages;
2  $S := \{k - 2^j + 1 \mid j = \log k, \ldots, 0\}$;
3  $W_i := [k - 2^{\log k - i + 1} + 1, k - 2^{\log k - i}]$ for $i = 1, \ldots, \log k$ and $W_{\log k + 1} = \{k\}$;
4  Choose $F \subseteq \{1, \ldots, k\}$ such that $|F \cap W_i| = \min\{f(i) - f(i-1), |W_i|\}$ for each $i$;
5  **foreach** *pagefault during the phase* **do**
6      **if** *it is arrival belonging to $F$* **then**  ask for new prediction $P$;
7      **if** *it is arrival belonging to $S$* **then**  synchronize with $P$;
8      **if** *requested page is still not in cache* **then**  evict random unmarked page;

9  Load all pages marked during the phase;
10  Switch to Follower (Algorithm 1);

---

Synchronization with $P$ (Line 7) works as follows. All pages previously evicted by random evictions return to the cache and the same number of pages not present in $P$ is evicted. We denote $E_i = E_i^- \cup E_i^+$ the set of pages evicted at the beginning of $W_i$, where pages in $E_i^-$ are requested during $W_i$ while those in $E_i^+$ are not. Note that algorithm's and predictor's cache may not become the same after the synchronization. Since the algorithm starts with pages in $M$ and loads only clean pages, we have the following observation.

**Observation 3.1.** Let $C_i$, $|C_i| = c_i$ be the set of clean pages arriving before the start of $W_i$. Then, $E_i \subseteq M \cup C_i$ and $|E_i| = |M \cup C_i| - k = c_i$.

We assume that the predictor is lazy and does not load pages that are not requested. Therefore, no page from $E_i^+$ will be loaded during $W_i$ by the predictor and the same holds for Robust, implying the following.

**Observation 3.2.** For every $i = 1, \ldots, \log k$, we have $E_i^+ \subseteq E_{i+1}$ and therefore $E_i \setminus E_{i+1} \subseteq E_i^-$.

Synchronization with the marking cache performed by Robust is to ensure that the difference between the cache of the algorithm and Belady can be bounded by costs incurred by Belady locally using Observation 2.2 instead of diverging over time solely due to incorrect predictions.

**Implementation suggestions.** Algorithms are described as to simplify the analysis. Synchronization in Robust (line 7) should be done lazily as to make use of the most recent prediction. At arrivals of clean pages, one may evict a page not present in predictor's cache instead of a random unmarked page; one can also ask for a fresh prediction (at most $2\,\mathrm{OPT}$ additional queries). The second synchronization with the marking cache in Robust (line 9) can be omitted. With $f(i) = 0$, one can query the predictor only at clean arrivals, using at most $2\,\mathrm{OPT}$ predictions in total. We recommend a lazy implementation. Since Robust is not 1-consistent, one may also switch from Follower only once Follower's cost is at least a constant (e.g. 2 or 3) times higher than the cost of Belady.

We denote $H_i$ the $i$th phase of $\mathrm{Robust}_f$ and $H_i^-$ a hypothetical marking phase which ends just before $H_i$ starts. Note that $H_i^-$ might overlap with $H_{i-1}$. But $H_1, H_2, \ldots$ are disjoint and we denote $G_{i,i+1}$ the time interval between the end of $H_i$ and the beginning of $H_{i+1}$. $c(H_i)$ is the number of clean pages during phase $H_i$. For a given time period $X$, we define $\Delta^A(X)$, $\Delta^B(X)$, and $\Delta^P(X)$ the costs incurred by F&R, Belady, and the predictor respectively during $X$ and $\eta(X)$ the error of predictions received during $X$.

Here is the main lemma about the performance of Robust. Overview of its proof is deferred to Section 3.1.

**Lemma 3.3.** *Denote $X_i = H_{i-1} \cup H_i^- \cup H_i$. During the phase $H_i$, $\mathrm{Robust}_f$ receives at most $f(\log k) + 1$ predictions and we have*

$$\mathbb{E}[\Delta^A(H_i)] \leq O(1) f^{-1} \left( \frac{\eta(H_i)}{\Delta^B(X_i)} \right) \Delta^B(X_i). \tag{1}$$

*At the same time, we also have*

$$\mathbb{E}[\Delta^A(H_i)] \leq O(\log k) \Delta^B(X_i) \text{ and} \tag{2}$$

$$\Delta^A(H_i) \leq O(k) + O(k)\eta(H_i). \tag{3}$$

The following lemma is useful to analyze the cost incurred during the Follower part of the algorithm. The proof of Theorem 1.1 then combines it with with Lemma 3.3 and can be found in the full version of the paper.

**Lemma 3.4.** *Consider the gap $G_{i,i+1}$ between phases $H_i$ and $H_{i+1}$ of $\mathrm{Robust}_f$. We have*

$$\Delta^A(G_{i,i+1}) \leq \Delta^B(G_{i,i+1}) + \Delta^B(H_i).$$

## 3.1 Analysis of $\mathrm{Robust}_f$

The full version of this section and the proof of Lemma 3.3 can be found in Appendix (Section 3.2), here we include a short overview. Charging a page fault on a page evicted due to predictor's advice to a single incorrect action prediction can only give us smoothness linear in the prediction error. This is in contrast with next-arrival predictions where algorithms can be analyzed by estimating lengths of eviction chains caused by each incorrect prediction, as proposed by Lykouris and Vassilvitskii (2021). To achieve sublinear smoothness, we need to charge each such page fault to a long interval of incorrect predictions. This is the most challenging part of our analysis because Belady also moves and the same prediction incorrect at one time step may be correct at another time step. We estimate the error of predictions received during each window by introducing window rank which bounds the prediction error from below accounting for the movements of Belady.

## 4 Well-separated queries to the predictor

The full version of this section, which can be found in Appendix, contains a consistent and smooth algorithm for MTS proving Theorem 1.3 and extends our analysis of F&R to the setting where the

queries to the predictor need to be separated by at least $a$ time steps, proving Theorem 1.5. In MTS, the cost functions usually do not satisfy any Lipschitz property. Therefore, the difference between the cost of the state reported by the predictor and the state of the optimal algorithm does not need to be proportional to their distance. We show that a state satisfying this property which is close to the predicted state can be found using a classical technique for design of algorithms for MTS called *work functions*, see (Chrobak and Larmore, 1996) for reference. Then, we use the approach of Emek et al. (2009) to interpolate between predictions received at times $t$ and $t + a$. In the case of caching, the performance of F&R in this regime is the same as if it has received $a$ incorrect predictions for each prediction error. Therefore, $\eta$ in its smoothness bound needs to be multiplied by $a$.

## 5 EXPERIMENTS

We perform an empirical evaluation of our caching algorithm F&R on the same datasets and with the same predictors as the previous works (Lykouris and Vassilvitskii, 2021; Antoniadis et al., 2023; Im et al., 2022). We use the following datasets.

- BrightKite dataset (Cho et al., 2011) contains data from a certain social network. We create a separate caching instance from the data of each user, interpreting check-in locations as pages. We use it with cache size $k = 10$ and choose instances corresponding to the first 100 users with the longest check-in sequences requiring at least 50 page faults in the optimal policy.
- CitiBike dataset contains data about bike trips in a bike sharing platform CitiBike. We create a caching instance from each month in 2017, interpreting starting stations of the trips as pages, and trimming length of each instance to 25 000. We use it with cache size $k = 100$.

Some of the algorithms in our comparison use next-arrival predictions while F&R uses action predictions that can be generated from next-arrival predictions. Therefore, we use predictors which predict the next arrival of the requested page and convert it to action predictions. This process was used and described by Antoniadis et al. (2023) and we use their implementation of the predictors. Our algorithm is then provided limited access to the resulting action predictions while the algorithm of Im et al. (2022) has limited access to the original next-arrival predictions.

- Synthetic predictions: compute the exact next arrival time computed from the data and add noise to this number. This noise comes from a log-normal distribution with the mean parameter $\mu = 0$ and the standard deviation parameter $\sigma$. We use $\sigma \in [0, 50]$.
- PLECO predictor proposed by Anderson et al. (2014): This model estimates the probability $p$ of a page being requested in the next time step and we interpret this as a prediction that the next arrival of this page will be in $1/p$ time steps. The model parameters were fitted to BrightKite dataset and not adjusted before use on CitiBike.
- POPU – a simple predictor used by Antoniadis et al. (2023): if a page appeared in $p$ fraction of the previous requests, we predict its next arrival in $1/p$ time steps.

In our comparison, we include the following algorithms: offline algorithm Belady which we use to compute the optimal number of page faults OPT, standard online algorithms LRU and Marker (Fiat et al., 1991), ML-augmented algorithms using next arrival predictions L&V (Lykouris and Vassilvitskii, 2021), LMark and LnonMark (Rohatgi, 2020), FtPM which, at each step, evicts an unmarked page with the furthest predicted next arrival time, and algorithms for action predictions FtP and T&D (Antoniadis et al., 2023). We use the implementation of all these algorithms published by Antoniadis et al. (2023). We implement algorithm AQ (Im et al., 2022) and our algorithm F&R.

**Notes on implementation of** F&R**.** We follow the recommendations in Section 3 except that Follower switches to Robust whenever its cost is $\alpha = 1$ times higher compared to Belady in the same period. With higher $\alpha$, the performance of F&R approaches FtP on the considered datasets. With $k = 10$ (BrightKite dataset), we use $F = [1, 6, 9]$ corresponding to $f(i) = i$. Note that, with such small $k$, polynomial and exponential $f$ would also give a very similar $F$. With $k = 100$ (CitiBike dataset), we use exponential $f(i) = 2^{i+1} - 1$. With $a$-separated queries, Follower uses LRU heuristic when prediction is unavailable, and Robust ignores $F$, querying the predictor at each page fault separated from the previous query by at least $a$ time steps.

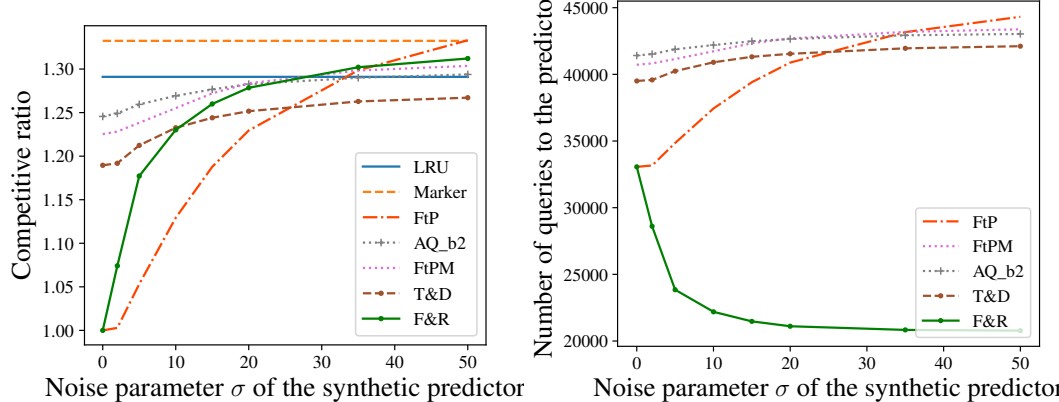

Figure 1: BrightKite dataset with Synthetic predictor, standard deviation at most 0.003 and 300 resp.

| Predictor | Marker | FtP | AQ_b8 | FtPM_a1 | FtPM_a5 | F&R_a1 | F&R_a5 | F&R_a20 |
|-----------|--------|-------|-------|---------|---------|--------|--------|---------|
| POPU | 1.861 | 1.739 | 1.782 | 1.776 | 1.833 | 1.800 | 1.802 | 1.803 |
| PLECO | 1.861 | 2.277 | 1.875 | 1.877 | 1.867 | 1.878 | 1.879 | 1.879 |

Figure 2: Competitive ratios on CitiBike dataset with $k = 100$, standard deviation at most 0.001

**Results.** Figures 1 and 2 contain averages of 10 independent experiments. Figure 1 shows that the performance of F&R with high-quality predictions is superior to the previous ML-augmented algorithms except for FtP which follows the predictions blindly and is also 1-consistent. With high $\sigma$, the performance of T&D becomes better. This is true also for F&R with $F = [1..10]$, suggesting that T&D might be more efficient in using erroneous predictions. The second plot shows the total number of times algorithms query the predictor over all instances. Response to such query is a single page missing from predictor's cache in the case of F&R and T&D and next arrival times of $b$ pages in the case of AQ_b. Note that FtPM is equivalent to the non-parsimonious version of AQ with $b = k$. F&R makes the smallest number of queries: with perfect predictions, it makes exactly OPT queries and this number decreases with higher $\sigma$ as F&R spends more time in Robust.

Figure 2 shows that F&R performs well in the regime with $a$-separated queries. While the performance of FtPM with POPU predictor worsens considerably towards Marker already with $a = 5$, F&R keeps its improvement over Marker even with $a = 20$. Predictions produced by PLECO seem much less precise as suggested by FtP with PLECO being worse than Marker and smaller number of such predictions either improves (AQ, FtPM) or does not affect performance (F&R) of considered algorithms. Further details of our experimental results are presented in Appendix (Section 5).

## 6 CONCLUSIONS

We present algorithms for MTS and caching with action predictions working in the setting where the number of queries or the frequency of querying the predictor are limited. We have shown that one can achieve theoretical as well as empirical performance comparable to the setting with unlimited access to the predictor, possibly enabling usage of precise but heavy-weight prediction models in environments with scarce computational resources.

## REPRODUCIBILITY STATEMENT

The appendix contains a full version of our paper which includes proof of all the theorems and lemmas. We provide textual description of the implementation of our algorithm in Section 5. The code of our implementation can be found at `https://github.com/marek-elias/caching/`

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

# FULL VERSION OF THE PAPER ALGORITHMS FOR CACHING AND MTS WITH REDUCED NUMBER OF PREDICTIONS

**Karim Abdel Sadek**
University of Amsterdam*
karim.abdel.sadek@student.uva.nl

**Marek Eliáš**
Department of Computing Sciences
Bocconi University
marek.elias@unibocconi.it

## ABSTRACT

ML-augmented algorithms utilize predictions to achieve performance beyond their worst-case bounds. Producing these predictions might be a costly operation – this motivated Im et al. (2022) to introduce the study of algorithms which use predictions parsimoniously. We design parsimonious algorithms for caching and MTS with *action predictions*, proposed by Antoniadis et al. (2023), focusing on the parameters of consistency (performance with perfect predictions) and smoothness (dependence of their performance on the prediction error). Our algorithm for caching is 1-consistent, robust, and its smoothness deteriorates with the decreasing number of available predictions. We propose an algorithm for general MTS whose consistency and smoothness both scale linearly with the decreasing number of predictions. Without the restriction on the number of available predictions, both algorithms match the earlier guarantees achieved by Antoniadis et al. (2023).

## 1 INTRODUCTION

Caching, introduced by Sleator and Tarjan (1985), is a fundamental problem in online computation important both in theory and practice. Here, we have a fast memory (cache) which can contain up to $k$ different pages and we receive a sequence of requests to pages in an online manner. Whenever a page is requested, it needs to be loaded in the cache. Therefore, if the requested page is already in the cache, it can be accessed at no cost. Otherwise, we suffer a *page fault*: we have to evict one page from the cache and load the requested page in its place. The page to evict is to be chosen without knowledge of the future requests and our target is to minimize the total number of page faults.

Caching is a special case of Metrical Task Systems introduced by Borodin et al. (1992) as a generalization of many fundamental online problems. In the beginning, we are given a metric space $M$ of states which can be interpreted as actions or configurations of some system. We start at a predefined state $x_0 \in M$. At time steps $t = 1, 2, \ldots$, we receive a cost function $\ell_t \colon M \to \mathbb{R}^+ \cup \{0, +\infty\}$ and we need to make a decision: either to stay at $x_{t-1}$ and pay a cost $\ell_t(x_{t-1})$, or to move to another, possibly cheaper state $x_t$ and pay $\ell_t(x_t) + d(x_{t-1}, x_t)$, where the distance $d(x_{t-1}, x_t)$ represents the transition cost between states $x_{t-1}$ and $x_t$.

The online nature of both caching and MTS forces an algorithm to make decisions without knowledge of the future which leads to very suboptimal results in the worst case (Borodin et al., 1992; Sleator and Tarjan, 1985). A recently emerging field of learning-augmented algorithms, introduced in seminal papers by Kraska et al. (2018) and Lykouris and Vassilvitskii (2021), investigates approaches to improve the performance of algorithms using predictions, possibly generated by some ML model. In general, no guarantee on the accuracy of these predictions is assumed. Therefore, the performance of learning-augmented algorithms is usually evaluated using the following three parameters:

*Consistency.* Performance with perfect predictions, preferably close to optimum.

---

*The presentation of this paper was financially supported by the Amsterdam ELLIS Unit and Qualcomm. Work completed while Abdel Sadek was in his final year of BSc at Bocconi University

*Robustness.* Performance with very bad predictions, preferably no worse than what is achievable by known algorithms which do not utilize predictions.

*Smoothness.* Algorithm's performance should deteriorate smoothly with increasing prediction error between the consistency and robustness bound.

These three parameters express a desire to design algorithms that work very well when receiving reasonably accurate predictions most of the time and, in the rest of the cases, still satisfy state-of-the-art worst-case guarantees. See the survey by Mitzenmacher and Vassilvitskii (2020) for more information.

Producing predictions is often a computationally intensive task, therefore it is interesting to understand the interplay between the number of available predictions and the achievable performance. In their inspiring work, Im et al. (2022) initiated the study of learning-augmented algorithms which use the predictions parsimoniously. In their work, they study caching with next-arrival-time predictions introduced by Lykouris and Vassilvitskii (2021). Their algorithm uses $O(b \log_{b+1} k)$ OPT predictions, where OPT is the number of page faults incurred by the offline optimal solution and $b \in \mathbb{N}$ is a parameter. It achieves smoothness linear in the prediction error. It satisfies tight consistency bounds: with perfect predictions, it incurs at most $O(\log_{b+1} k)$ OPT page faults and no algorithm can do better. In other words, it achieves a constant competitive ratio with unrestricted access to predictions ($b = k$) and, with $b$ a small constant, its competitive ratio deteriorates to $O(\log k)$ which is comparable to the best competitive ratio achievable without predictions. One of their open questions is whether a similar result could be proved for MTS.

In this paper, we study parsimonious algorithms for MTS working with *action predictions* which were introduced by Antoniadis et al. (2023). Here, each prediction describes the state of an optimal algorithm at the given time step and its error is defined as the distance from the actual state of the optimal algorithm. The total prediction error is the sum of errors of the individual predictions. In the case of caching, action predictions have a very concise representation, see Section 2.1. Unlike next-arrival-time predictions, action predictions can be used for any MTS. Using the method of Blum and Burch (2000), it is easy to achieve near-optimal robustness for any MTS losing only a factor $(1 + \epsilon)$ in consistency and smoothness. Therefore, we study how the reduced number of predictions affects the consistency and smoothness parameters. We consider the following two regimes.

*Bounded number of predictions:* The algorithm can request a prediction whenever it prefers as far as the total number of requested predictions is bounded by $b\,\mathrm{OPT}$, where $b$ is a parameter. This regime is similar to Im et al. (2022).

*Well-separated queries to the predictor:* The queries to the predictor need to be separated by at least $a$ time steps, for some parameter $a$. This captures the situation when producing each prediction takes more than one time step.

## 1.1 OUR RESULTS

We evaluate the algorithm's performance using *competitive ratio* which is, roughly speaking, the worst-case ratio between the cost incurred by the algorithm and the cost of the offline optimum, see Section 2 for a formal definition. We say that an algorithm achieves consistency $\alpha$ and robustness $\beta$ if its competitive ratio is at most $\alpha$ when provided with perfect predictions and at most $\beta$ with arbitrarily bad predictions. For a given function $g$, we call an algorithm $g(\eta)$-smooth if its competitive ratio is at most $g(\eta)$ whenever provided with predictions with the total error at most $\eta$.

Our first contribution is an algorithm for caching which receives action predictions describing the states of the optimal offline algorithm $\mathrm{Belady}$ proposed by Belady (1966). High quality such predictor based on imitation learning was already designed by Liu et al. (2020). Its empirical evaluation within existing algorithms designed for action predictions was performed by Chledowski et al. (2021).

**Theorem 1.1.** *Let $f$ be an increasing convex function such that $f(0) = 0$ and $f(i) \le 2^i - 1$ for each $i \ge 0$. There is an algorithm for caching requiring $O(f(\log k))$ OPT predictions which achieves consistency* 1*, robustness $O(\log k)$, and smoothness $O(f^{-1}(\eta/OPT))$, where $\eta$ denotes the total prediction error with respect to* $\mathrm{Belady}$ *and* OPT *is the number of page faults of* $\mathrm{Belady}$.

In fact, the number of required predictions is slightly smaller than what is stated in the theorem. Table 1 shows numbers of predictions and achieved smoothness for some natural choices of $f$. Already with

Table 1: Smoothness vs. number of predictions.

| $f(i)$ | # of predictions | smoothness |
|---|---|---|
| $2^i - 1$ | $O(\sqrt{k})\,\mathrm{OPT}$ | $O(1 + \log(\frac{\eta}{\mathrm{OPT}} + 1))$ |
| $i^2$ | $O(\log^2 k)\,\mathrm{OPT}$ | $O(\sqrt{2\frac{\eta}{\mathrm{OPT}}})$ |
| $i$ | $O(\log k)\,\mathrm{OPT}$ | $O(\frac{\eta}{\mathrm{OPT}})$ |
| $0$ | $2\,\mathrm{OPT}$ | $O(\frac{k\eta}{\mathrm{OPT}})$ |

$O(\sqrt{k})\,\mathrm{OPT}$ predictions, our bounds are comparable to Antoniadis et al. (2023) whose algorithm asks for a prediction in every step, its consistency is constant and its smoothness is logarithmic in $\eta$. The algorithm also works with $f(i) = 0$. In that case, it asks for at most $2\,\mathrm{OPT}$ predictions and still remains 1-consistent. However, its smoothness is not very good. We use sliding marking phases and a careful distribution of queries of the predictor over the time horizon. This allows us to avoid dealing with so called "ancient" pages considered by Rohatgi (2020) and Antoniadis et al. (2023), resulting in an algorithm with better consistency and a simpler analysis.

We discuss tightness of our bounds in Section 7. We show that with, for example, only $0.5OPT$ available predictions, no algorithm can be better than $O(\log k)$-competitive – a guarantee comparable to the best classical online algorithms without predictions. We also show that the number of predictions used by our algorithm is close to optimal.

**Theorem 1.2.** *Let $f$ be an increasing function. Any $f(\eta)$-smooth algorithm for caching with action predictions, i.e., an algorithm whose competitive ratio with predictions of error $\eta$ is $f^{-1}(\eta)$ for any $\eta > 0$, has to use at least $f(\ln k)\,\mathrm{OPT}$ predictions.*

For general MTS, we cannot bound the number of used predictions as a function of OPT. The reason is that any instance of MTS can be scaled to make OPT arbitrarily small, allowing us to use only very few predictions. We propose an algorithm which queries the predictor once in every $a$ time steps, making at most $T/a$ queries in total, where $T$ denotes the length of the input sequence.

**Theorem 1.3.** *There is a deterministic algorithm for any MTS which receives a prediction only once per each $a$ time steps and its cost is at most $O(a) \cdot (\mathrm{OFF} + 2\eta)$, where $\mathrm{OFF}$ denotes the cost of an arbitrary offline algorithm and $\eta$ the error of predictions with respect to this algorithm.*

This is a more general statement than Theorem 1.1 which requires OFF to be Belady. Considering any offline optimal algorithm OFF, Theorem 1.3 implies a smoothness $O(a) \cdot (1 + 2\eta/\mathrm{OPT})$ and consistency $O(a)$. Our algorithm is based on work functions. For $a = 1$, its smoothness is $1 + 2\eta/\mathrm{OFF}$, see Section 4, which improves upon the smoothness bound of $1 + 4\eta/\mathrm{OFF}$ by Antoniadis et al. (2023). It is not robust on its own. However, it can be combined with any online algorithm for the given MTS using the result of Blum and Burch (2000) achieving robustness comparable to that algorithm and losing only a factor of $(1 + \epsilon)$ in smoothness and consistency.

No algorithm receiving a prediction only once in $a$ time steps can be $o(a)$-consistent. This follows from the work of Emek et al. (2009) on advice complexity, see Section 7 for more details. The same can be shown for smoothness by modifying the lower bound construction of Antoniadis et al. (2023).

**Theorem 1.4.** *There is no $o(a\eta/\mathrm{OPT})$-smooth algorithm for MTS with action predictions which receives predictions only once in $a$ time steps.*

We can modify our algorithm for caching to ensure that the moments when the predictions are queried are separated by at least $a$ time steps, not losing too much of its performance.

**Theorem 1.5.** *There is an algorithm for caching which receives prediction at most once in $a \le k$ time steps and using at most $O(f(\log k))\,\mathrm{OPT}$ predictions in total which is $O(1)$-consistent, $O(\log k)$-robust and $O(f^{-1}(a\eta/\mathrm{OPT}))$-smooth.*

In Section 5, we provide empirical results suggesting that our algorithm's performance can be comparable to the performance of algorithms imposing no limitations on their use of predictions. Our algorithm may therefore be useful especially with heavy-weight predictors like (Liu et al., 2020).

In Section 8, we provide an algorithm for an alternative prediction setup which we call FitF oracle: each prediction says which of the pages in the current algorithms cache will be requested furthest in the future.

## 1.2 RELATED WORK

The most related work is by Im et al. (2022), who studied caching with next arrival time predictions. A smaller number of predictions affects the consistency of their algorithm: with $b(\log k/\log b)$ OPT predictions, they achieve consistency $O(\log k/\log b)$ and they show that this is tight. They also show that their algorithm achieves linear smoothness. In contrast, our algorithm is 1-consistent when receiving at least OPT predictions. This demonstrates that action predictions, although not containing more bits, seem to contain useful information about the input instance in a more condensed form. See (Antoniadis et al., 2023) for comparison and connections between these prediction setups. Drygala et al. (2023) study ski rental and bahncard problems with predictions of a fixed cost.

There are several other papers on caching with predictions, including (Lykouris and Vassilvitskii, 2021; Rohatgi, 2020; Wei, 2020; Emek et al., 2009; Antoniadis et al., 2023; 2022) which design algorithms asking for a prediction at each time step. Consistency parameters achieved by these algorithms are constants greater than 1. Note that those using black-box methods to achieve robustness are $(1+\epsilon)$-consistent (e.g. (Wei, 2020)). We can explicitly compare our smoothness bounds to Antoniadis et al. (2023) who use the same kind of predictions: their smoothness is $O(1+\log(\frac{\eta}{\text{OPT}}+1))$ with unlimited use of predictions while our algorithm achieves the same smoothness bound with $O(\sqrt{k})$ OPT predictions. We compare the smoothness of the other algorithms experimentally in Section 5. Antoniadis et al. (2022) study a prediction setup where each prediction is only a single bit, however their algorithms need to receive it in every time step. Gupta et al. (2022) study several problems including caching in a setting where each prediction is correct with a constant probability.

Antoniadis et al. (2023) proposed a 1-consistent and $(1+4\eta/\text{OPT})$-smooth algorithm for MTS with action predictions which can be robustified by loosing factor $(1+\epsilon)$ in consistency and smoothness. Getting smoothness bounds sublinear in $\eta/\text{OPT}$ for specific MTS problems other than caching remains a challenging open problem even with unlimited number of predictions and this holds even for weighted caching. Specific results on weighted caching are by Jiang et al. (2022) who studied it in a setup requiring very verbose predictions and by Bansal et al. (2022) whose bounds depend on the number of weight classes. There is also a consistency/robustness trade-off by Lindermayr et al. (2022) for $k$-server.

Since the seminal papers by Kraska et al. (2018) and Lykouris and Vassilvitskii (2021) which initiated the study of learning-augmented algorithms, many computational problems were considered.

There are papers on ski rental (Purohit et al., 2018; Antoniadis et al., 2021), secretary and matching problems (Dütting et al., 2021; Antoniadis et al., 2020), online-knapsack (Im et al., 2021; Zeynali et al., 2021; Boyar et al., 2022), graph exploration (Eberle et al., 2022), online TSP (Bernardini et al., 2022), energy efficient scheduling (Bamas et al., 2020), flow-time scheduling (Azar et al., 2021; 2022), restricted assignment (Lattanzi et al., 2020), non-clairvoyant scheduling Purohit et al. (2018); Lindermayr and Megow (2022), and online page migration (Indyk et al., 2022). In offline setting, there is a work of Dinitz et al. (Dinitz et al., 2021) on matching, Chen et al. (Chen et al., 2022) on graph algorithms, Polak and Zub (Polak and Zub, 2022) on flows, Sakaue and Oki (Sakaue and Oki, 2022) on discrete optimization problems, and Ergun et al. (Ergun et al., 2022) on clustering. We also refer to (Lindermayr and Megow, 2022) to an updated list of results in the area.

There are numerous works on advice complexity of online problems, where algorithms are given certain number of bits of information about the future which are guaranteed to be correct. This is different from our setting, where we receive predictions of unknown quality. We refer to the survey by Boyar et al. (2017), work of Dobrev et al. (2009) on caching, Emek et al. (2009) on MTS, and further papers by Hromkovič et al. (2010); Böckenhauer et al. (2017).

There are already works on predictors for caching. Jain and Lin (2016) proposed a binary classifier called Hawkey which predicts which pages will be kept in cache by $\mathrm{Belady}$, providing us with action predictions. Their result was later improved by Shi et al. (2019) who designed a model called Glider for the same classification problem. There is a very precise model by Liu et al. (2020) whose main output can be interpreted as an action prediction although it has a second prediction head which

produces next arrival predictions. This model is large and relatively slow and served as a motivation for this work. Chledowski et al. (2021) evaluated the performance of existing ML-augmented algorithms with this predictor.

## 2 PRELIMINARIES

Consider an algorithm ALG for MTS which produces a solution $x_0, x_1, \ldots, x_T$ for an instance $I$ consisting of cost functions $\ell_1, \ldots, \ell_T$. We denote $\text{cost}(\text{ALG}(I)) = \sum_{t=1}^{T} (\ell_t(x_t) + d(x_{t-1}, x_t))$. We say that ALG is $r$-competitive with respect to an offline algorithm OFF if there is an absolute constant $\alpha \in \mathbb{R}$ such that $\mathbb{E}[\text{cost}(\text{ALG}(I))] \leq r \cdot \text{cost}(\text{OFF}(I)) + \alpha$ for any instance $I$. If ALG is $r$-competitive with respect to an optimal offline algorithm, we say that ALG is $r$-competitive and call $r$ the competitive ratio of ALG. In the classical setting (without predictions), the best achievable competitive ratios are $\Theta(\log k)$ for caching (Fiat et al., 1991) and of order $poly \log n$ for MTS (Bartal et al., 2006; Bubeck et al., 2019), where $n$ is the number of points in the underlying metric space $M$. We refer to (Borodin and El-Yaniv, 1998) for a textbook treatment.

### 2.1 ACTION PREDICTIONS FOR MTS

Antoniadis et al. (2023) proposed a prediction setup which they call *action predictions*, where the predictions tell us what a good algorithm would do. More precisely, at each time $t$, the algorithm receives a prediction $p_t$ of a state where some offline algorithm OFF moves to at time $t$. The error of prediction $p_t$ is then $\eta_t = d(p_t, o_t)$, where $o_t$ is the real state of OFF at time $t$. The total prediction error is defined as $\eta = \sum_{t=1}^{T} \eta_t$.

Considering the case of caching, the state corresponds to a cache content, and the prediction error is the number of pages present in the cache of OFF and absent from the predicted cache content. A whole cache content may seem like a huge piece of information, but action predictions for caching can be implemented in a very succinct way. Antoniadis et al. (2023) explain how to represent them with only $O(\log k)$ bits per time step when they are received at each time step. Our algorithm asks, in each query, a specific number of indices of pages which are present in its cache but absent from the predicted cache. When we talk about a bound on the number of provided predictions, this bound applies both to the number of such queries as well as to the total number of indices reported by the predictor during the running time of the algorithm. There are predictors which can generate predictions of a similar kind by Jain and Lin (2016); Shi et al. (2019); Liu et al. (2020). See (Antoniadis et al., 2023) for a detailed treatment of this prediction setup and a comparison to other setups for caching.

### 2.2 CACHING: BELADY'S ALGORITHM, MARKING, AND LAZY ALGORITHMS

The classical optimal offline algorithm for caching proposed by Belady (1966) is denoted Belady in this paper. At each page fault, it evicts a page which is going to be requested furthest in the future. In the case of a tie, i.e., if there are several pages in the cache which will not be requested anymore, it chooses one of them arbitrarily. Our caching algorithm assumes that the predictor is trying to simulate Belady. The following useful property allows us to detect errors in the predictions quickly. It was recently used by Eliáš et al. (2024).

**Observation 2.1.** Consider request sequence $r_1, \ldots, r_T$. For any $t \leq T$, the cost incurred by Belady for $r_1, \ldots, r_T$ until time $t$ is the same as the cost of Belady with input $r_1, \ldots, r_t$.

To see this, it is enough to note that the solution produced by Belady with input $r_1, \ldots, r_T$ agrees until time $t$ with the solution produced by Belady on $r_1, \ldots, r_t$ which breaks ties based on the arrival times in $r_{t+1}, \ldots, r_T$.

We use properties of *marking* algorithms in this work. Such algorithms split the input sequence into phases, i.e., maximal subsequences where at most $k$ distinct pages are requested. Usually, the first phase starts in the beginning and the next phase follows just after the end of the previous one. However, we will consider phases starting at arbitrary moments. Let $O$ be the cache content of an algorithm in the beginning of the phase. Whenever a page is requested for the first time during the phase, we call this moment an *arrival* and we *mark* the page. At the end of the phase, the set

$M$ of marked pages will have size $k$: some of them belong to $O$ and are called *old* while those in $C = M \setminus O$ are called *clean*. Exactly $|C|$ pages from $O$ remain unmarked until the end of the phase.

Marking algorithms is a class of algorithms which never evict a marked page and all of them have cache content $M$ at the end of the phase. Belady is not marking and our algorithm is not marking either, although it uses ideas from marking to achieve desired robustness and smoothness properties. At the end of each phase, we can bound the difference between the cache content of some algorithm and marking.

**Observation 2.2.** Let $c$ be the cost incurred by some algorithm during a marking phase. Then, $c \geq |M \setminus S|$, where $S$ is the cache content of the algorithm at the end of the phase and $M$ is the set of $k$ pages requested during the phase.

This is because each page in $p \in M$ has to be present in algorithm's cache when requested during the phase. If $p \notin S$, then the algorithm must have evicted it during the phase incurring cost 1.

**Observation 2.3.** If a page $p$ is evicted by Belady at time $t$, then $p$ is not going to be requested in the marking phase containing $t$ anymore.

If $p$ is evicted by Belady at time $t$, then the currently requested page $r_t$ and $k - 1$ pages from the cache are $k$ distinct pages that are requested before the moment when $p$ is requested next time. The current phase then needs to end before that moment.

We say that an algorithm is *lazy* if it evicts only one page at a time and only at a page fault. Belady is lazy while our algorithm, as described, may not be. However, any algorithm can be made lazy without increasing its cost. See (Borodin and El-Yaniv, 1998) for more information about caching.

**Observation 2.4.** The difference in the cache content of two lazy algorithms can increase only if both of them have a page fault. In that case, it can increase by at most 1.

### 2.3 MTS AND ADVICE COMPLEXITY

Advice complexity studies the performance of algorithms depending on the number of bits of precise information about the instance available in advance. In the case of MTS, Emek et al. (2009) study the situation when algorithm receives $\frac{1}{a} \log n$ bits of information about the state of some optimal offline algorithm, being able to identify its true state once in $a$ time steps. They formulate the following proposition for OFF being an optimal algorithm, but the proof does not use its optimality and it can be any algorithm located at $q_i$ at time $ia$.

**Proposition 2.5** (Emek et al. (2009)). *There is an algorithm which, with knowledge of state $q_i$ of algorithm* OFF *at time $ia$ for $i = 1, \ldots, T/a$, is $O(a)$-competitive w.r.t.* OFF.

In our context, we can say that the algorithm from the preceding proposition is $O(a)$-consistent if $q_1, \ldots, q_{T/a}$ are states of an optimal solution. However, it is not smooth because it may not be possible to relate the cost of OFF to the prediction error with respect to OPT.

## 3 BOUNDED NUMBER OF PREDICTIONS

In this section, we prove Theorem 1.1. We propose an algorithm called F&R which consists of two parts: Follower and Robust. It starts with Follower which is 1-consistent, but lacks in smoothness and robustness. At each page fault, Follower recomputes Belady for the part of the sequence seen so far and checks whether it also has a page fault. If yes, it copies the behavior of the predictor (Line 3). Otherwise, it must have received an incorrect prediction before. Therefore, it switches to Robust (Line 5) which is no more 1-consistent, but achieves desired smoothness and robustness. Robust runs for one marking phase and then returns back to Follower. At such moment, the predictor's and the algorithm's cache can be very different and Follower may need to lazily synchronize with the predictor (Line 4).

Algorithm Robust runs during a single marking phase starting at the same moment, splitting it into windows as follows (assuming $k$ is a power of 2): The first window $W_1$ starts at the beginning of the phase and lasts $k/2$ arrivals, i.e., it ends just before the arrival number $k/2 + 1$. $W_i$ follows the $W_{i-1}$ and its length is half of the remaining arrivals in the phase. The last window $W_{\log k + 1} = \{k\}$ lasts until the end of the phase. Robust comes with an increasing convex function $f$ such that $f(0) = 0$

---

**Algorithm 1:** Follower

---

1  $P :=$ initial cache content;                            `// Prediction for time 0`
2  **foreach** *pagefault* **do**
3     **if** $r_t \notin P$ *and* Belady *has a pagefault* **then**  query new prediction $P$ and evict any $p \in C \setminus P$;
4     **else if** $r_t \in P$ **then**  evict arbitrary $p \notin P$;
5     **else**  Switch to Robust (Algorithm 2);

---

and $f(i) \le 2^j - 1$. Faster growing $f$ does not further improve our smoothness bounds. Function $f$ determines that we should request $f(i) - f(i-1)$ predictions in window $i$. If the window is too small, we ask for prediction at each time step. Robust starts with the cache content of a marking algorithm whose new phase would start at the same moment (Line 1). In the case of a page fault, it evicts an unmarked page chosen uniformly at random. At arrivals belonging to the sets $S$ and $F$, it performs synchronization with the predictor and queries the predictor's state respectively. The synchronization is always performed with respect to the most recent prediction $P$ which, in the case of lazy (or lazified) predictors, implicitly incorporates information from the previous predictions.

---

**Algorithm 2:** Robust$_f$ (one phase)

---

1  Load $k$ distinct most recently requested pages;
2  $S := \{k - 2^j + 1 \mid j = \log k, \ldots, 0\}$;
3  $W_i := [k - 2^{\log k - i + 1} + 1, k - 2^{\log k - i}]$ for $i = 1, \ldots, \log k$ and $W_{\log k + 1} = \{k\}$;
4  Choose $F \subseteq \{1, \ldots, k\}$ such that $|F \cap W_i| = \min\{f(i) - f(i-1), |W_i|\}$ for each $i$;
5  **foreach** *pagefault during the phase* **do**
6     **if** *it is arrival belonging to* $F$ **then**  ask for new prediction $P$;
7     **if** *it is arrival belonging to* $S$ **then**  synchronize with $P$;
8     **if** *requested page is still not in cache* **then**  evict random unmarked page;
9  Load all pages marked during the phase;
10  Switch to Follower (Algorithm 1);

---

Synchronization with $P$ (Line 7) works as follows. All pages previously evicted by random evictions return to the cache and the same number of pages not present in $P$ is evicted. We denote $E_i = E_i^- \cup E_i^+$ the set of pages evicted at the beginning of $W_i$, where pages in $E_i^-$ are requested during $W_i$ while those in $E_i^+$ are not. Note that algorithm's and predictor's cache may not become the same after the synchronization. Since the algorithm starts with pages in $M$ and loads only clean pages, we have the following observation.

**Observation 3.1.** Let $C_i, |C_i| = c_i$ be the set of clean pages arriving before the start of $W_i$. Then, $E_i \subseteq M \cup C_i$ and $|E_i| = |M \cup C_i| - k = c_i$.

We assume that the predictor is lazy and does not load pages that are not requested. Therefore, no page from $E_i^+$ will be loaded during $W_i$ by the predictor and the same holds for Robust, implying the following.

**Observation 3.2.** For every $i = 1, \ldots, \log k$, we have $E_i^+ \subseteq E_{i+1}$ and therefore $E_i \setminus E_{i+1} \subseteq E_i^-$.

Synchronization with the marking cache performed by Robust is to ensure that the difference between the cache of the algorithm and Belady can be bounded by costs incurred by Belady locally using Observation 2.2 instead of diverging over time solely due to incorrect predictions.

**Implementation suggestions.** Algorithms are described as to simplify the analysis. Synchronization in Robust (line 7) should be done lazily as to make use of the most recent prediction. At arrivals of clean pages, one may evict a page not present in predictor's cache instead of a random unmarked page; one can also ask for a fresh prediction (at most $2\,\text{OPT}$ additional queries). The second synchronization with the marking cache in Robust (line 9) can be omitted. With $f(i) = 0$, one can query the predictor only at clean arrivals, using at most $2\,\text{OPT}$ predictions in total. We recommend a lazy implementation. Since Robust is not 1-consistent, one may also switch from Follower only once Follower's cost is at least a constant (e.g. 2 or 3) times higher than the cost of Belady.

We denote $H_i$ the $i$th phase of $\mathrm{Robust}_f$ and $H_i^-$ a hypothetical marking phase which ends just before $H_i$ starts. Note that $H_i^-$ might overlap with $H_{i-1}$. But $H_1, H_2, \ldots$ are disjoint and we denote $G_{i,i+1}$ the time interval between the end of $H_i$ and the beginning of $H_{i+1}$. $c(H_i)$ is the number of clean pages during phase $H_i$. For a given time period $X$, we define $\Delta^A(X)$, $\Delta^B(X)$, and $\Delta^P(X)$ the costs incurred by F&R, Belady, and the predictor respectively during $X$ and $\eta(X)$ the error of predictions received during $X$.

Here is the main lemma about the performance of Robust. Its proof is deferred to Section 3.2.

**Lemma 3.3.** *Denote $X_i = H_{i-1} \cup H_i^- \cup H_i$. During the phase $H_i$, $\mathrm{Robust}_f$ receives at most $f(\log k) + 1$ predictions and we have*

$$\mathbb{E}[\Delta^A(H_i)] \le O(1)f^{-1}\left(\frac{\eta(H_i)}{\Delta^B(X_i)}\right)\Delta^B(X_i). \tag{1}$$

*At the same time, we also have*

$$\mathbb{E}[\Delta^A(H_i)] \le O(\log k)\Delta^B(X_i) \text{ and} \tag{2}$$

$$\Delta^A(H_i) \le O(k) + O(k)\eta(H_i). \tag{3}$$

### 3.1 Analysis of Follower

**Lemma 3.4.** *Consider the gap $G_{i,i+1}$ between phases $H_i$ and $H_{i+1}$ of $\mathrm{Robust}_f$. We have*

$$\Delta^A(G_{i,i+1}) \le \Delta^B(G_{i,i+1}) + \Delta^B(H_i).$$

*Proof.* There are $\Delta^B(G_{i,i+1})$ page faults served at line 3 because Belady also has those page faults. To bound the cost incurred on line 4, we denote $P, B, M$ the cache contents of the predictor, Belady, and Robust respectively at the end of the phase $H_i$. The synchronization with $P$ costs at most $|(P \setminus M) \cap B|$ if we omit costs incurred by Belady which were already accounted for above. However, $(P \setminus M) \cap B = (B \setminus M) \cap P \subseteq B \setminus M$ and $|B \setminus M| \le \Delta^B(H_i)$ by Observation 2.2. $\square$

*Proof of Theorem 1.1.* The cost of Follower until the start of $H_1$ is the same as the cost of Belady. Therefore, by lemmas 3.4 and 3.3 equation 1, the cost of F&R, in expectation, is at most

$$OPT + \sum_i \Delta^B(H_i) + \sum_i O(1)\Delta^B(X_i)f^{-1}\left(\frac{\eta(H_i)}{\Delta^B(X_i)}\right),$$

where the sums are over all phases of Robust and $X_i = H_{i-1} \cup H_i^- \cup H_i$. Since phases $H_i$ are disjoint and the same holds for $H_i^-$, this expression is at most $OPT \cdot O(f^{-1}(\eta/OPT))$ by concavity of $f^{-1}$, implying the smoothness bound for F&R.

If we use bound equation 2 instead of equation 1, we get $O(\log k)\, OPT$ – the robustness bound. Since there must be at least one error during the execution of Follower to trigger each execution of Robust, equation 3 implies that the cost of F&R is at most $OPT + \eta O(k)$. With $\eta = 0$, this implies 1-consistency of F&R. Follower queries the predictor only at a page fault by OPT and the prediction consists of a single page evicted by the predictor. Robust may ask for up to $f(\log k) + 1$ predictions in each phase, each of them consisting of indices of at most $c(H_i)$ pages from F&R cache not present in the predictor's cache. This gives both $O(f(\log k))\, OPT$ queries to the predictor as well as $O(f(\log k))\, OPT$ predicted indices in total. $\square$

Note that $\mathrm{Robust}_f$ can rarely use full $O(f(\log k))$ predictions, because the last windows are not long enough. More precise calculation of numbers of predictions can be found in Appendix A.

### 3.2 Analysis of $\mathrm{Robust}_f$

First, we relate the number of clean pages in a robust phase to the costs incurred by Belady.

**Observation 3.5.** Consider a phase $H$ denoting $C(H)$ the set of clean pages arriving during $H$. We have

$$c(H) := |C(H)| \le \Delta^B(H^-) + \Delta^B(H).$$

*Proof.* There are $k + c(H)$ pages requested during $H^- \cup H$. Therefore, any algorithm, and Belady in particular, has to pay cost $\Delta^B(H^- \cup H) \geq c(H)$. □

**Lemma 3.6.** *Consider phase $H_i$. Cost incurred by* Robust *for synchronization with marking in Line 1 is at most $\Delta^B(H_{i-1} \cup H_i^-)$.*

*Proof.* Let $M$ denote the $k$ distinct most recently requested pages – these are marked pages during $H_i^-$. We consider two cases.

If $H_i^- \cap H_{i-1} = \emptyset$, then whole $H_i^-$ was served by Follower. Each $p \in M$ must have been in the cache of both Follower and $P$ when requested and Follower would evict it afterwards only if $P$ did the same. Therefore, Robust needs to load at most $\Delta^P(H_i^-) = \Delta^B(H_i^-)$ pages.

If $H_i^-$ and $H_{i-1}$ overlap, let $M'$ denote the set of pages marked during $H_{i-1}$. At the end of $H_{i-1}$, Robust loads $M'$ and Follower loads only pages from $M$ until the end of $H_i^-$. Therefore, Robust starting $H_i$ needs to evict only pages from $M' \setminus M$. Now, note that there are $|M' \cup M|$ distinct pages requested during $H_{i-1} \cup H_i^-$ and therefore $|M' \setminus M| = |M' \cup M| - k \leq \Delta^B(H_{i-1} \cup H_i^-)$. □

We consider costs incurred by Robust during window $W_i$ for $i = 1, \ldots, \log k + 1$. Note that $E_1 = E_1^+ = E_1^- = \emptyset$, since $W_1$ starts at the beginning of the phase and there are no clean pages arriving strictly before $W_1$.

**Lemma 3.7.** *Expected cost incurred by* Robust$_f$ *during $W_1$ is at most $2c_2$. For $i = 2, \ldots, \log k + 1$, we have*

$$\mathbb{E}[\Delta^A(W_i)] \leq |E_{i-1}^-| + c_i - c_{i-1} + 2(|E_i^-| + c_{i+1} - c_i),$$

*denoting $c_{\log k+2} = c(H)$.*

*Proof.* First, consider the costs during $W_1$. There are $c_2 - c_1 = c_2$ clean pages arriving during $W_1$ and Robust has a page fault due to each of them, evicting a random unmarked page. In the worst case, all these clean pages arrive at the beginning of $W_1$. Therefore, at arrival $c_2 + 1$, there are $c_2$ pages evicted which were chosen among unmarked pages uniformly at random. Let $U_a$ denote the set of unmarked pages at arrival $a$. We have $U_1 = M$ (the initial cache content of Robust) and none of those pages get marked during first $c_2$ arrivals. During every arrival $a = c_2 + 1, \ldots, k/2 = S[2] - 1$, a single unmarked page is marked and we have $|U_a| = k - (a - c_2)$. As in the classical analysis of Marker (see (Borodin and El-Yaniv, 1998) and references therein), the probability of the requested unmarked page being evicted is $c_2/|U_a|$. We have

$$\Delta^A(W_1) = c_2 + \sum_{a=c_2+1}^{S[2]-1} \frac{c_2}{|U_a|} \leq c_2 + \frac{k}{2} \cdot \frac{c_2}{k/2} = 2c_2.$$

For $i \geq 2$, there are $c_i$ pages evicted before the start of $W_i$: those in $E_{i-1}^+$ were evicted due to synchronization with the predictor and the rest were evicted in randomized evictions – those are loaded back to the cache at the beginning of $W_i$, causing cost $c_i - |E_{i-1}^+| = |E_{i-1}^-| + c_i - c_{i-1}$. After this synchronization, all unmarked pages are in the cache except those belonging to $E_i$.

During $W_i$, pages from $E_i^-$ and $c_{i+1} - c_i$ new clean pages are requested causing page faults which are resolved by evicting a random unmarked page from the cache. In the worst case, these page faults all happen in the beginning of the window, leaving more time for page faults on randomly-evicted pages. Let $\bar{a}$ denote the first arrival after these page faults and $U_{\bar{a}}$ the set of unmarked pages at that moment. At arrival $\bar{a}$, there are $c_{i+1}$ pages missing from the cache: pages from $E_i^+$ which are not going to be requested during $W_i$ and $|E_i^-| + c_{i+1} - c_i = c_{i+1} - |E_i^+|$ unmarked pages were chosen uniformly at random from $U_{\bar{a}} \setminus E_i = U_{\bar{a}} \setminus E_i^+$. This is because only pages which were marked since the beginning of $W_i$ until $\bar{a}$ are those from $E_i^-$ and they were not present in the cache before they got marked. We compute the expected number of page faults on the randomly evicted pages. Since they are unmarked when evicted, such page faults can happen only on arrivals.

At arrival $a$, the set of unmarked pages $U_a$ has size $k - (a - c_{i+1})$. For any $a \in W_i$ such that $a \geq \bar{a}$, we have $U_a \cap E_i^- = \emptyset$ and pages in $U_a \cap E_i^+$ are evicted with probability 1. So, $c_{i+1} - |E_i^+|$ evicted

pages are picked uniformly at random from $U_a \setminus E_i^+$ of size at least $k - (a - c_{i+1}) - |E_i^+|$. Therefore, our expected cost is at most

$$\sum_{a=\bar{a}}^{S[i+1]-1} \frac{c_{i+1} - |E_i^+|}{k - a + c_{i+1} - |E_i^+|} \leq \sum_{a \in W_i} \frac{c_{i+1} - |E_i^+|}{k - a} \leq \frac{k}{2^i} \cdot \frac{c_{i+1} - |E_i^+|}{k/2^i} = c_{i+1} - |E_i^+|$$

which is equal to $|E_{i-1}^-| + c_i - c_{i-1}$. Note that $k - a \geq k - S[i+1] - 1 = 2^{\log k - i} = k/2^i$. $\qquad \square$

For $i = 1, \dots, \log k$, we define

$$\text{rank}(W_i) := |E_{i+1} \cap B_{i+1}| - \Delta^B(W_i),$$

where $B_i$ denotes the cache content of Belady at the beginning of $W_i$. We do not define $\text{rank}(W_i)$ for $i = 1 + \log k$. Later, we relate rank to the prediction error. We have the following lemma.

**Lemma 3.8.** *During a phase $H$, the expected cost of $\text{Robust}_f$ is at most*

$$3 \sum_{i=1}^{\log k + 1} \text{rank}(W_{i-1}) + 3c(H) + 6\Delta^B(H).$$

*Proof.* First, we observe that

$$|E_i^-| \leq \text{rank}(W_{i-1}) + \Delta^B(W_{i-1}) + \Delta^B(W_i) \tag{4}$$

holds for $i = 2, \dots, \log k + 1$. This is because $|E_i^- \cap B_i| \leq |E_i \cap B_i| = \text{rank}(W_{i-1}) + \Delta^B(W_{i-1})$ and $|E_i^- \setminus B_i| \leq \Delta^B(W_i)$ due to pages from $E_i^-$ being requested during $W_i$ and Belady having to load them. Combining equation 4 with Lemma 3.7, and summing over all windows, we get the statement of the lemma. $\qquad \square$

Now, we relate the rank of a window to the prediction error.

**Lemma 3.9.** *Denote $\eta_i$ the error of predictions arriving during $W_i$. We have*

$$\eta(W_i) \geq |F \cap W_i| \text{rank}(W_i).$$

*Proof.* Prediction error at time $t$ is $\eta_t = |B_t \setminus P_t|$. At the end of $W_i$, it is at least $|E_{i+1} \cap B_{i+1}|$. Due to laziness of the predictor, $|B_t \setminus P_t|$ can increase only if both predictor and Belady have a page fault: in that case it may increase by 1, see Observation 2.4. Therefore, at any time $t$ during $W_i$, we have $\eta_t = |B_t \setminus P_t| \geq |E_{i+1} \cap F_{i+1}| - \Delta^B(W_i) = \text{rank}(W_i)$. Since we query the predictor at arrivals belonging to $F$, the total error of received predictions is at least $|F \cap W_i| \text{rank}(W_i)$. $\qquad \square$

We will analyze intervals of windows starting when some particular incorrect prediction was introduced and ending once it was corrected. The following lemma charging the increase of rank to the arriving clean pages and costs incurred by Belady will be used to bound the number of such intervals.

**Lemma 3.10.** *For $i = 1, \dots, \log k$, we have*

$$\text{rank}(W_i) - \text{rank}(W_{i-1}) \leq \Delta^B(W_{i-1}) + c_{i+1} - c_i,$$

*denoting $W_0$ an empty window before $W_1$ with $\text{rank}(W_0) = 0$.*

*Proof.* It is enough to show that

$$|E_{i+1} \cap B_{i+1}| \leq |E_i \cap B_i| + \Delta^B(W_i) + c_{i+1} - c_i.$$

Since the right-hand side is always non-negative, we only need to consider the case when the left-hand side is positive. We show how to charge pages in $E_{i+1} \cap B_{i+1}$ either to pages in $E_i \cap B_i$ or to $\Delta^B(W_i)$ and $c_{i+1} - c_i$.

Since $|E_i| = c_i \leq c_{i+1} = |E_{i+1}|$, we can construct an injective map $\beta \colon E_i \to E_{i+1}$, such that $\beta(p) = p$ for each $p \in E_i \cap E_{i+1}$. There are $|E_{i+1}| - |E_i| = c_{i+1} - c_i$ pages $p$ such that $\beta^{-1}(p)$ is not defined. We show that, for each $p \in E_{i+1} \cap B_{i+1}$ for which it is defined, $\beta^{-1}(p)$ is either a page in $E_i \cap B_i$ or a page loaded by Belady during $W_i$. There are two cases.

- $p \in E_i \cap E_{i+1}$ implying $\beta^{-1}(p) = p$. Then either $p \in B_i$ and therefore $p \in E_i \cap B_i$, or $p \in B_{i+1} \setminus B_i$ implying that Belady has loaded $p$ during $W_i$.

- $p \notin E_i \cap E_{i+1}$ implying $q = \beta^{-1}(p) \in E_i \setminus E_{i+1}$. By Observation 3.2, $q \in E_i^-$, i.e., it must have been requested during $W_i$. If $q \in B_i$ then, $q \in E_i \cap B_i$. Otherwise, Belady must have loaded $q$ during $W_i$.

To sum up: $\beta$ is an injective map and $\beta^{-1}(p)$ does not exist for at most $c_{i+1} - c_i$ pages $p \in E_{i+1} \cap B_{i+1}$. All other $p \in E_{i+1} \cap B_{i+1}$ are mapped by $\beta^{-1}$ to a unique page either belonging to $E_i \cap B_i$ or loaded by Belady during $W_i$. $\qquad\square$

*Proof of Lemma 3.3.* $\mathrm{Robust}_f$ receives a prediction only at an arrival belonging to $F$. Since $F$ contains $\sum_{i=1}^{\log k+1} |F \cap W_i| \leq \sum_{i=1}^{\log k}(f(i) - f(i-1)) + |W_{\log k+1}| \leq f(\log k) + 1$ arrivals, because $|W_{\log k+1}| = 1$ and $f(0) = 0$, there are at most $f(\log k) + 1$ queries to the predictor.

To prove equations (1,2,3), we always start with bounds proved in lemmas 3.5, 3.8 and 3.6. In the rest of the proof, we write $H$ instead of $H_i$ and $X$ instead of $H_i$ to simplify the notation.

To get equation equation 2, note that $\mathrm{rank}(W_i) \leq c(H)$ for each window $i$. Therefore, by lemmas 3.5, 3.8 and 3.6, we can bound $\mathbb{E}[\Delta^A(H)]$ by

$$\Delta^B(X) + 3c(H)\log k + 3c(H) + 6\Delta^B(H) \leq O(\log k)\Delta^B(X).$$

To get equation equation 3, note that $\Delta^B(H) \leq k$, $c(H) \leq k$, and $\mathrm{rank}(W_i) > 0$ only if $\eta_i \geq |E_{i+1} \cap B_{i+1}| > 1$. Therefore, we get $\Delta^A(H) \leq O(k) + \eta(H)O(k)$.

Now we prove equation equation 1. We define $Q_m = \{i \mid \mathrm{rank}(W_i) < m \text{ and } \mathrm{rank}(W_{i+1}) \geq m\}$ and $Q = \sum_{m=1}^{k} |Q_m|$. We can bound $Q$ using Lemma 3.10. We have

$$Q = \sum_{m=1}^{k} |Q_m| = \sum_{i=1}^{\log k} \max\{0, \mathrm{rank}(W_i) - \mathrm{rank}(W_{i-1})\} \leq \sum_{i=1}^{\log k} (2\Delta^B(W_i) + c_{i+1} - c_i), \quad (5)$$

which is equal to $2\Delta^B(H) + c(H)$.

We bound $\sum_{j=1}^{\log k} \mathrm{rank}(W_j)$ as a function of $Q$ and $\eta(H)$

$$\sum_{i=1}^{\log k} \mathrm{rank}(W_i) \leq 2Q \cdot f^{-1}\left(\frac{a\eta(H)}{Q}\right). \qquad (6)$$

This relation is proved in Proposition 3.11 with $a = 1$ and, together with equation 5, gives us the desired bound

$$\mathbb{E}[\Delta^A(H)] \leq O(1)\Delta^B(H)f^{-1}\left(\frac{a\eta(H)}{Q}\right). \qquad (7)$$

$\qquad\square$

**Proposition 3.11.**
$$\sum_{i=1}^{\log k} \mathrm{rank}(W_i) \leq 2Q \cdot f^{-1}\left(\frac{\eta(H)}{Q}\right).$$

*Proof.* We rearrange the sum of ranks in the following way. We define $L_m = \{i \mid \mathrm{rank}(W_i) \geq m\}$, and $a_{i,m}$, such that $L_m = \bigcup_{i \in Q_m}(i, i + a_{i,m}]$ for each $m$. We can write

$$\sum_{i=1}^{\log k} \mathrm{rank}(W_i) = \sum_{m=1}^{k} |L_m| = \sum_{m=1}^{k} \sum_{i \in Q_m} a_{i,m}. \qquad (8)$$

On the other hand, we can write $\eta_i \geq \sum_{m=1}^{\mathrm{rank}(W_i)} |F \cap W_i|$ (Lemma 3.9) which allows us to decompose the total prediction error $\eta(H)$ as follows:

$$\eta(H) \geq \sum_{m=1}^{k} \sum_{i \in L_m} |F \cap W_i| = \sum_{m=1}^{k} \sum_{i \in Q_m} \sum_{j=1}^{a_{i,m}} |F \cap W_{i+j}|.$$

Let $i^*$ denote the first window such that $|W_{i^*}| < f(i^*) - f(i^* - 1)$. If $i + a_{i,m} < i^*$, then $\sum_{j=1}^{a_{i,m}} |F \cap W_{i+j}| = f(i + a_{i,m}) - f(i) \geq f(a_{i,m})$ by convexity of $f$. If this is not the case, we claim that $\sum_{j=1}^{a_{i,m}} |F \cap W_{i+j}| \geq f(a_{i,m}/2)$. This is clearly true if $i + \lceil a_{i,m}/2 \rceil < i^*$. Otherwise, we have $\sum_{j=1}^{a_{i,m}} |F \cap W_{i+j}| \geq \sum_{j=\lceil a_{i,m}/2 \rceil}^{a_{i,m}} |F \cap W_{i+j}| \geq 2^{a_{i,m}/2}$ because $i + a_{i,m} \leq k$ and therefore $|W_{i+\lceil a_{i,m}/2 \rceil}| \geq 2^{a_{i,m}/2}$. By our assumptions about $f$, we have $f(a_{i,m}/2) \leq 2^{a_{i,m}/2}$.

So, we have the following lower bound on $\eta(H)$:

$$\eta(H) \geq \sum_{m=1}^{k} \sum_{i \in Q_m} f(a_{i,m}/2). \tag{9}$$

By convexity of $f$, this lower bound is smallest if all $a_i^m$ are the same, i.e., equal to the total rank divided by $Q = \sum_m |Q_m|$ and then $f(a_{i,m}/2) = \eta(H)/Q$ for each $i$ and $m$. Combining equation 8 and equation 9, we get

$$\sum_{i=1}^{\log k} \frac{\mathrm{rank}(W_i)}{2} = \sum_{m=1}^{k} \sum_{i \in Q_m} f^{-1}\big(f\big(\frac{a_{i,m}}{2}\big)\big) \leq Q \cdot f^{-1}\big(\frac{\eta(H)}{Q}\big). \qquad \square$$

## 4 WELL-SEPARATED QUERIES TO THE PREDICTOR

### 4.1 MTS

We consider the setting when we are able to receive a prediction once every $a$ time steps, for some parameter $a \in \mathbb{N}$. Without loss of generality, we assume that $T$ is a multiple of $a$. In time steps $ia$, where $i = 1, \ldots, T/a$, we receive a prediction $p_i \in M$. State $p_i$ itself might be very bad, e.g. $\ell_{ia}(p_i)$ might be infinite. We use work functions to see whether there is a more suitable point nearby $p_i$.

**Work functions.** Consider an MTS on a metric space $M$ with a starting state $x_0 \in M$ and a sequence of cost functions $\ell_1, \ldots, \ell_T$. For each time step $t = 1, \ldots, T$ and state $x \in M$, we define a *work function* as

$$w_t(x) = \min \Big\{ d(y_t, x) + \sum_{i=1}^{t} d(y_{i-1}, y_i) + \ell_i(y_i) \Big\},$$

where the minimum is taken over all $y_0, \ldots, y_t$ such that $y_0 = x_0$. In other words, it is the cheapest way to serve all the cost functions up to time $t$ and end in state $x$. Work function is a major tool for design of algorithms for MTS and satisfies the following property.

**Observation 4.1.** For any $x, y \in M$ and any time $t$, we have $w_t(x) \leq w_t(y) + d(y, x)$.

This holds because one way to serve the cost functions $\ell_1, \ldots, \ell_t$ is to follow the best solution which ends in state $y$ and then move to $x$. If $w_t(p_i) = w_t(y) + d(y, p_i)$, we can see that $p_i$ is not a very good state, since the best solution ending in $p_i$ goes via $y$. We say that $p_i$ is *supported* by $y$.

**Algorithm of Emek et al. (2009).** Algorithm 3 was proposed by Emek et al. (2009) in the context of advice complexity. It receives the state of an offline optimum algorithm every $a$ time steps.

### 4.2 ALGORITHM FTSP

Given $q_{i-1}$ and $\ell_t$ for $t = (i-1)a + 1, \ldots, ia$, we define

$$\mathrm{wf}_i(x) = \min \Big\{ d(x_{ia}, x) + \sum_{j=(i-1)a+1}^{ia} d(x_{j-1}, x_j) + \ell_j(x_j) \Big\},$$

---

**Algorithm 3:** One cycle of algorithm by Emek et al. (2009)

---

1 $q_i :=$ reported state of OFF at time $ia$;
2 $j := 0, c := 0$;
3 **for** $t = ia + 1, \ldots, (i + 1)a$ **do**
4     $x'_t := \arg\min_{x \in B(q_i, 2^j)}\{d(x, x_{t-1}) + \ell_t(x)\}$;
5     **while** $c + d(x, x'_t) + \ell_t(x'_t) > 2^j$ **do**
6         $j := 2 * j$;
7         $x'_t := \arg\min_{x \in B(q_i, 2^j)}\{d(x, x_{t-1}) + \ell_t(x)\}$;
8     $x_t := x'_t, c := c + d(x, x_t) + \ell_t(x_t)$;

---

where the minimum is taken over $x_{(i-1)\alpha}, \ldots x_{i\alpha} \in M$ such that $x_{(i-1)a} = q_{i-1}$. In fact, it is the work function at the end of an MTS instance with initial state $q_{i-1}$ and request sequence $\ell_t$ for $t = (i-1)a + 1, \ldots, ia$. Instead of $p_i$, we choose point

$$q_i = \arg\min_{x \in M}\left\{ \text{wf}_i(x) \;\middle|\; \text{wf}_i(x) = \text{wf}_i(p_i) - d(x, p_i) \right\},$$

i.e., the "cheapest" state supporting $p_i$ in $\text{wf}_i$. After computing $q_i$, we run one cycle of Algorithm 3. This algorithm which we call "Follow the Scarce Predictions" (FtSP), is summarized in Algorithm 4.

---

**Algorithm 4:** FtSP

---

1 **for** $i = 0, \ldots, T/a$ **do**
2     receive prediction $p_i$;
3     use $p_i$ to compute $q_i$;
4     run one cycle of Algorithm 3 starting at $q_i$;

---

Let $Q$ denote the best (offline) algorithm which is located at $q_i$ at time step $ia$ for each $i = 1, \ldots, T/a$. We have

$$\text{cost}(Q) = \sum_{i=1}^{T/a} \text{wf}_i(q_i).$$

We can relate the cost of $Q$ to the prediction error using the following lemma. Together with Proposition 2.5, it gives a bound

$$\text{cost}(\text{ALG}) \leq O(a)(\text{OFF} + 2\eta),$$

implying Theorem 1.3.

**Lemma 4.2.** *Let* OFF *be an arbitrary offline algorithm and $o_i$ denote its state at time $ia$ for $i = 1, \ldots, T/a$. If $Q$ was computed from predictions $p_1, \ldots, p_{T/a}$, we have*

$$\text{cost}(Q) \leq \text{OFF} + 2\eta,$$

*where $\eta = \sum_{i=1}^{T/a} d(p_i, o_i)$ is the prediction error with respect to* OFF.

*Proof.* Denote $A_i$ an algorithm which follows the steps of $Q$ until time $ia$ and then follows the steps of OFF. We have

$$\text{cost}(A_i) \leq \text{cost}(A_{i-1}) + \text{wf}_i(q_i) - \text{wf}_i(o_i) + d(q_i, o_i)$$

because both $A_i$ and $A_{i-1}$ are at $q_{i-1}$ at time $(i-1)a$, and $A_{i-1}$ then travels to $o_i$ paying $\text{wf}_i(o_i)$ while $A_i$ travels to $q_i$ at $ia$ paying $\text{wf}_i(q_i)$ and its costs after $ia$ will be by at most $d(q_i, o_i)$ larger than the costs of $A_{i-1}$.

By Observation 4.1 and the choice of $q_i$, we have

$$\text{wf}_i(o_i) \geq \text{wf}_i(p_i) - d(o_i, p_i) = \text{wf}_i(q_i) + d(q_i, p_i) - d(o_i, p_i).$$

Combining the two preceding inequalities, we get

$$
\begin{aligned}
\mathrm{cost}(A_i) &\leq \mathrm{cost}(A_{i-1}) + \mathrm{wf}_i(q_i) - \mathrm{wf}_i(q_i) \\
&\quad + d(o_i, p_i) - d(p_i, q_i) + d(q_i, o_i) \\
&\leq \mathrm{cost}(A_{i-1}) + 2d(o_i, p_i),
\end{aligned}
$$

where the last step follows from the triangle inequality.

Since $\mathrm{OFF} = A_0$ and $Q = A_{T/a}$, we have

$$
Q \leq \mathrm{OFF} + 2 \sum_{i=1}^{T/a} d(o_i, p_i) = \mathrm{OFF} + 2\eta. \qquad \square
$$

## 4.3 CACHING

Follower cannot maintain 1-consistency in this setting. For the sake of theoretical bound, we can do the following: We serve the whole input sequence by subsequent phases of $\mathrm{Robust}_f$ which is $O(1)$-consistent with $F$ chosen in such a way that the arrivals in $F$ are separated by at least $a$. We prove the following replacement of Proposition 3.11.

**Proposition 4.3.**

$$
\sum_{i=1}^{\log k} \mathrm{rank}(W_i) \leq 2Q \cdot f^{-1}\left(\frac{a\,\eta(H)}{Q}\right).
$$

*Proof.* We rearrange the sum of ranks in the following way. We define $L_m = \{i \mid \mathrm{rank}(W_i) \geq m\}$, $Q_m = \{i \mid \mathrm{rank}(W_i) < m \text{ and } \mathrm{rank}(W_{i+1}) \geq m\}$, and $a_{i,m}$, such that $L_m = \bigcup_{i \in Q_m}(i, i + a_{i,m}]$ for each $m$. We can write

$$
\sum_{i=1}^{\log k} \mathrm{rank}(W_i) = \sum_{m=1}^{k} |L_m| = \sum_{m=1}^{k} \sum_{i \in Q_m} a_{i,m}. \tag{10}
$$

On the other hand, we can write $\eta_i \geq \sum_{m=1}^{\mathrm{rank}(W_i)} |F \cap W_i|$ (Lemma 3.9) which allows us to decompose the total prediction error $\eta(H)$ as follows:

$$
\eta(H) \geq \sum_{m=1}^{k} \sum_{i \in L_m} |F \cap W_i| = \sum_{m=1}^{k} \sum_{i \in Q_m} \sum_{j=1}^{a_{i,m}} |F \cap W_{i+j}|.
$$

Let $i^*$ denote the first window such that $\frac{|W_{i^*}|}{a} < f(i^*) - f(i^* - 1)$. If $i + a_{i,m} < i^*$, then $\sum_{j=1}^{a_{i,m}} |F \cap W_{i+j}| = f(i + a_{i,m}) - f(i) \geq f(a_{i,m})$ by convexity of $f$. If this is not the case, we claim that $\sum_{j=1}^{a_{i,m}} |F \cap W_{i+j}| \geq a^{-1} f(a_{i,m}/2)$.

- If $i + \lceil a_{i,m}/2 \rceil < i^*$: we have

$$
\sum_{j=1}^{a_{i,m}} |F \cap W_{i+j}| \geq \sum_{j=1}^{\lceil a_{i,m}/2 \rceil} |F \cap W_{i+j}| \geq f(a_{i,m}/2).
$$

- Otherwise: we have

$$
\sum_{j=1}^{a_{i,m}} |F \cap W_{i+j}| \geq \sum_{j=\lceil a_{i,m}/2 \rceil}^{a_{i,m}} |F \cap W_{i+j}| \geq \sum_{j=\lceil a_{i,m}/2 \rceil}^{a_{i,m}} \frac{1}{a}|W_{i+j}| \geq \frac{1}{a} 2^{a_{i,m}/2} \geq a^{-1} f(a_{i,m}/2)
$$

By our assumptions about $f$ saying that $f(a_{i,m}/2) \leq 2^{a_{i,m}/2}$.

So, we have the following lower bound on $\eta(H)$:

$$a\eta(H) \geq \sum_{m=1}^{k} \sum_{i \in Q_m} f(a_{i,m}/2). \tag{11}$$

By convexity of $f$, this lower bound is smallest if all $a_i^m$ are the same, i.e., equal to $a\eta(H)$ divided by $Q = \sum_m |Q_m|$ and then $f(a_{i,m}/2) = a\eta(H)/Q$ for each $i$ and $m$. Combining equation 10 and equation 11, we get

$$\sum_{i=1}^{\log k} \frac{\mathrm{rank}(W_i)}{2} = \sum_{m=1}^{k} \sum_{i \in Q_m} f^{-1}\big(f\big(\frac{a_{i,m}}{2}\big)\big) \leq Q \cdot f^{-1}\big(\frac{a\eta(H)}{Q}\big) \qquad \square$$

Using the proposition above in Equation equation 6 in the proof of Lemma 3.3 gives us the following smoothness bound:

**Lemma 4.4.** *Denote $X_i = H_{i-1} \cup H_i^- \cup H_i$. During the phase $H_i$, $\mathrm{Robust}_f$ receiving at most one prediction in $a$ time steps incurs the cost*

$$\mathbb{E}[\Delta^A(H_i)] \leq O(1) f^{-1}\left(\frac{a\eta(H_i)}{\Delta^B(X_i)}\right) \Delta^B(X_i).$$

Theorem 1.5 follows from summation of the bound above over all phases of $\mathrm{Robust}$ and concavity of $f^{-1}$, as in proof of Theorem 1.1.

## 5 EXPERIMENTS

We perform an empirical evaluation of our caching algorithm F&R on the same datasets and with the same predictors as the previous works (Lykouris and Vassilvitskii, 2021; Antoniadis et al., 2023; Im et al., 2022). We use the following datasets.

- BrightKite dataset (Cho et al., 2011) contains data from a certain social network. We create a separate caching instance from the data of each user, interpreting check-in locations as pages. We use it with cache size $k = 10$ and choose instances corresponding to the first 100 users with the longest check-in sequences requiring at least 50 page faults in the optimal policy.

- CitiBike dataset contains data about bike trips in a bike sharing platform CitiBike. We create a caching instance from each month in 2017, interpreting starting stations of the trips as pages, and trimming length of each instance to 25 000. We use it with cache size $k = 100$.

Some of the algorithms in our comparison use next-arrival predictions while F&R uses action predictions that can be generated from next-arrival predictions. Therefore, we use predictors which predict the next arrival of the requested page and convert it to action predictions. This process was used and described by Antoniadis et al. (2023) and we use their implementation of the predictors. Our algorithm is then provided limited access to the resulting action predictions while the algorithm of Im et al. (2022) has limited access to the original next-arrival predictions.

- Synthetic predictions: compute the exact next arrival time computed from the data and add noise to this number. This noise comes from a log-normal distribution with the mean parameter $\mu = 0$ and the standard deviation parameter $\sigma$. We use $\sigma \in [0, 50]$.

- PLECO predictor proposed by Anderson et al. (2014): This model estimates the probability $p$ of a page being requested in the next time step and we interpret this as a prediction that the next arrival of this page will be in $1/p$ time steps. The model parameters were fitted to BrightKite dataset and not adjusted before use on CitiBike.

- POPU – a simple predictor used by Antoniadis et al. (2023): if a page appeared in $p$ fraction of the previous requests, we predict its next arrival in $1/p$ time steps.

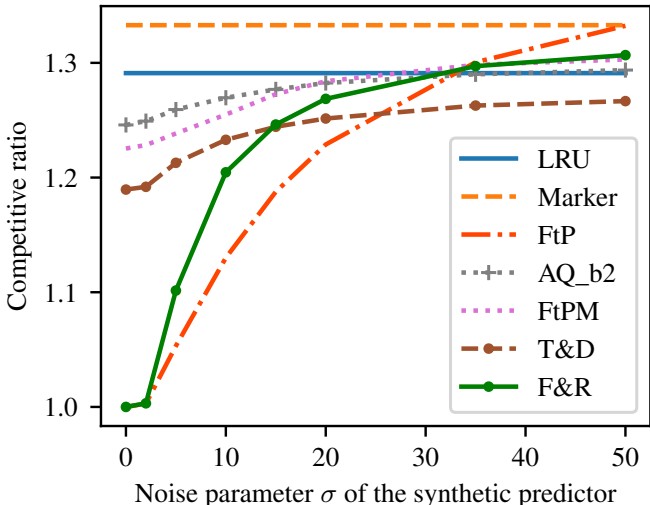

Figure 1: BrightKite dataset with Synthetic predictor: competitive ratio

In our comparison, we include the following algorithms: offline algorithm Belady which we use to compute the optimal number of page faults OPT, standard online algorithms LRU and Marker (Fiat et al., 1991), ML-augmented algorithms using next arrival predictions L&V (Lykouris and Vassilvit-skii, 2021), LMark and LnonMark (Rohatgi, 2020), FtPM which, at each step, evicts an unmarked page with the furthest predicted next arrival time, and algorithms for action predictions FtP and T&D (Antoniadis et al., 2023). We use the implementation of all these algorithms published by Antoniadis et al. (2023). We implement algorithm AQ (Im et al., 2022) and our algorithm F&R.

**Notes on implementation of** F&R**.** We follow the recommendations in Section 3 except that Follower switches to Robust whenever its cost is $\alpha = 1$ times higher compared to Belady in the same period. With higher $\alpha$, the performance of F&R approaches FtP on the considered datasets. With $k = 10$ (BrightKite dataset), we use $F = [1, 6, 9]$ corresponding to $f(i) = i$. Note that, with such small $k$, polynomial and exponential $f$ would also give a very similar $F$. With $k = 100$ (CitiBike dataset), we use exponential $f(i) = 2^{i+1} - 1$. With $a$-separated queries, Follower uses LRU heuristic when prediction is unavailable, and Robust ignores $F$, querying the predictor at each page fault separated from the previous query by at least $a$ time steps.

**Results.** Figures 1 and 3 contain averages of 10 independent experiments. Figure 1 shows that the performance of F&R with high-quality predictions is superior to the previous ML-augmented algorithms except for FtP which follows the predictions blindly and is also 1-consistent. With high $\sigma$, the performance of T&D becomes better. This is true also for F&R with $F = [1..10]$, suggesting that T&D might be more efficient in using erroneous predictions. Figure 2 shows the total number of times algorithms query the predictor over all instances. Response to such query is a single page missing from predictor's cache in the case of F&R and T&D and next arrival times of $b$ pages in the case of AQ_$b$. Note that FtPM is equivalent to the non-parsimonious version of AQ with $b = k$. F&R makes the smallest number of queries: with perfect predictions, it makes exactly OPT queries and this number decreases with higher $\sigma$ as F&R spends more time in Robust.

Figure 3 shows that F&R performs well in regime with $a$-separated queries. While the performance of FtPM with POPU predictor worsens considerably towards Marker already with $a = 5$, the performance of F&R worsens only very slowly. On CitiBike dataset, it keeps its improvement over Marker even with $a = 20$ (note that we use $k = 100$ with this dataset). Predictions produced by PLECO seem much less precise as suggested by FtP with PLECO being worse than Marker and smaller number of such predictions either improves (AQ, FtPM) or does not affect performance (F&R) of considered algorithms.

Figures 4 and 5 complements the comparison of F&R to existing ML-augmented algorithms for paging by including those omitted in Figure 1. With smaller $\sigma$, it again demonstrates the better

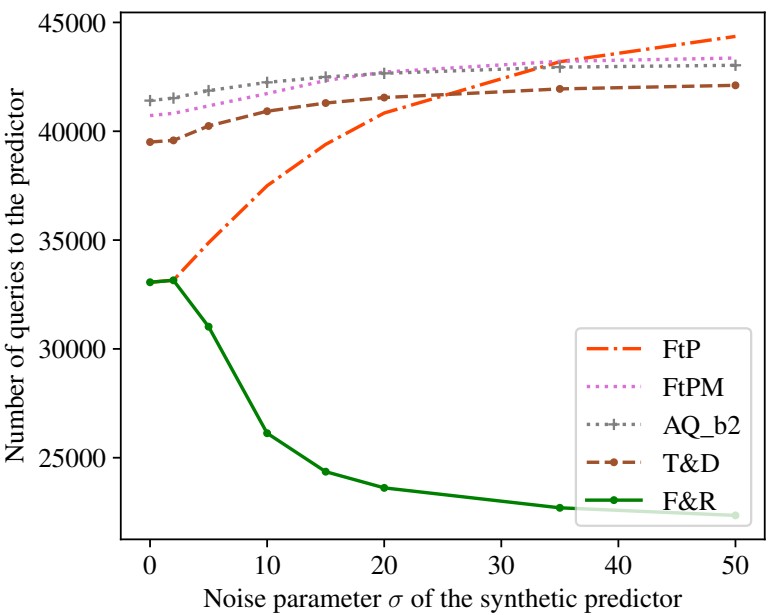

Figure 2: BrightKite dataset with Synthetic predictor: number of used predictors

| Dataset | Predictor | Marker | F&R_a1 | F&R_a2 | F&R_a3 | F&R_a5 | F&R_a8 | F&R_a20 |
|---------|-----------|--------|--------|--------|--------|--------|--------|---------|
| CitiBike | POPU | 1.862 | 1.800 | 1.802 | 1.802 | 1.802 | 1.803 | 1.803 |
| CitiBike | PLECO | 1.862 | 1.878 | 1.878 | 1.878 | 1.879 | 1.879 | 1.879 |
| BrightKite | POPU | 1.333 | 1.320 | 1.328 | 1.332 | 1.336 | 1.337 | 1.341 |
| BrightKite | PLECO | 1.333 | 1.371 | 1.374 | 1.376 | 1.377 | 1.378 | 1.378 |

| Dataset | Predictor | T&D | FtP | FtPM_a1 | FtPM_a5 | AQ_b8 | L&V | LMark | LnonMark |
|---------|-----------|-----|-----|---------|---------|-------|-----|-------|----------|
| CitiBike | POPU | 1.776 | 1.739 | 1.776 | 1.833 | 1.782 | 1.776 | 1.780 | 1.771 |
| CitiBike | PLECO | 1.847 | 2.277 | 1.877 | 1.866 | 1.875 | 1.877 | 1.876 | 1.863 |
| BrightKite | POPU | 1.276 | 1.707 | 1.262 | 1.306 | 1.263 | 1.262 | 1.264 | 1.266 |
| BrightKite | PLECO | 1.292 | 2.081 | 1.341 | 1.337 | 1.342 | 1.340 | 1.337 | 1.333 |

Figure 3: Competitive ratios with predictors POPU and PLECO

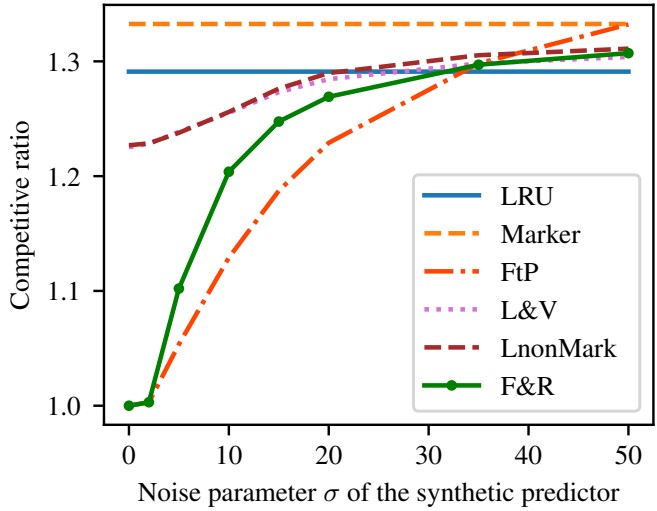

Figure 4: BrightKite dataset with Synthetic predictor: competitive ratio

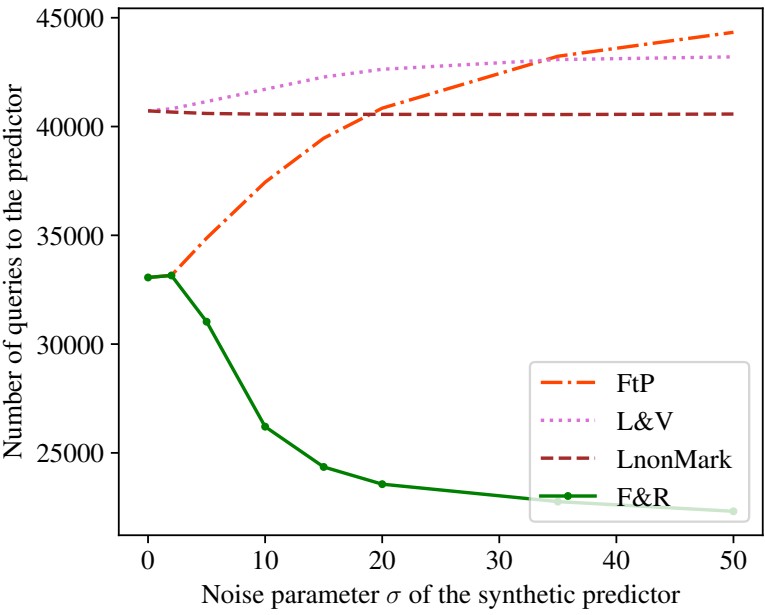

Figure 5: BrightKite dataset with Synthetic predictor: number of used predictions

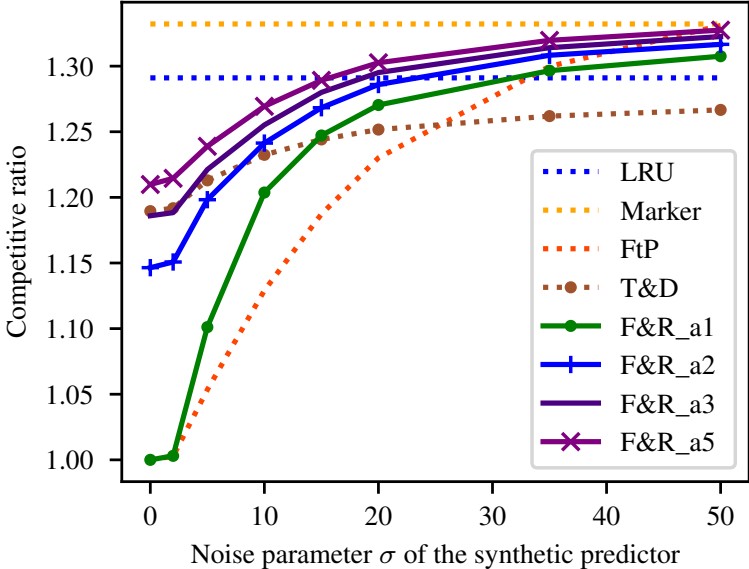

Figure 6: BrightKite dataset with Synthetic predictor: competitive ratio

consistency of F&R. With higher $\sigma$, F&R achieves performance comparable to both L&V and LnonMark, while using a smaller number of predictions. We have decided not to include LMark because its performance as well as number of predictions used were almost the same as of LnonMark. Note that, in the case of both algorithms, the number of used predictions is equal to the number of clean arrivals and therefore it does not change with the prediction error.

Figures 6 and 7 shows performance of F&R in regime with $a$-separated queries for different values of $a$. It shows a significant loss of consistency already with $a = 2$ compared to $a = 1$. However, with higher noise parameter $\sigma$, the difference in performance does not seem large. In this regime, the focus is on the gap between predictor queries rather than the total number of queries: F&R queries a predictor at each page fault separated from previous query by at least $a$ time steps. However, we decided to include also the plot of the total number of queries (Figure 7) because it shows that with $\sigma > 20$, F&R with $a = 1$ uses a smaller number of predictions than with $a = 2$ and even $a = 3$, while maintaining a better performance. This suggests that the freedom to choose the right moment for a query might be more important for the performance than the total number of used predictions.

Figure 8 shows experiments with a probabilistic predictor on the BrightKite dataset. In this setting, we consider a predictor that evicts the page requested furthest in the future with a given probability $1 - p$. On the other hand, it evicts a random page with probability $p$. The horizontal axis corresponds to the probability $p$. We can observe that better consistency of our algorithm compared to T&D is visible for $p$ up to $0.4$.

Each plot and table contains averages of 10 independent experiments. We have seen standard deviations at most 0.004 in the case of figures 1, 4, 6; 0.0015 for Figure 3 on CitiBike dataset and 0.0025 on BrightKite dataset, and 300 for figures 2, 5, 7, counting numbers of used predictions.

## 6 Conclusions

We present algorithms for MTS and caching with action predictions working in the setting where the number of queries or the frequency of querying the predictor are limited. We have shown that one can achieve theoretical as well as empirical performance comparable to the setting with unlimited access to the predictor, possibly enabling usage of precise but heavy-weight prediction models in environments with scarce computational resources.

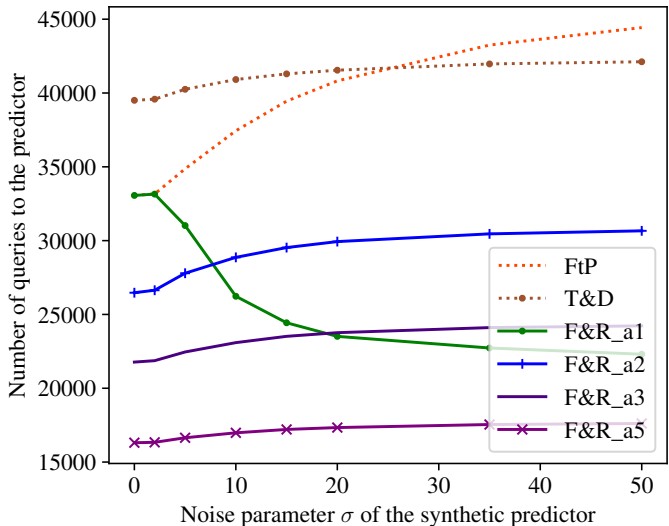

Figure 7: BrightKite dataset with Synthetic predictor: number of used predictions

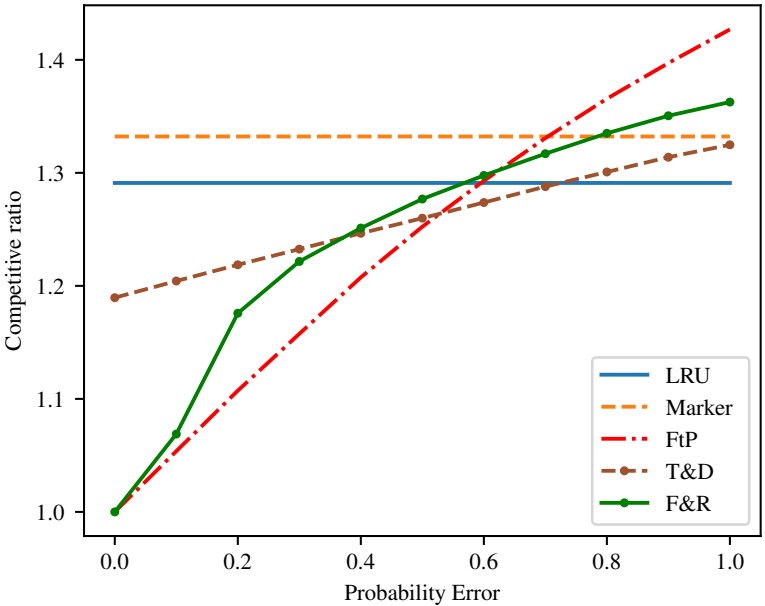

Figure 8: BrightKite dataset with probabilistic predictor: competitive ratio

# 7 LOWER BOUNDS

## 7.1 CACHING

Proof of the following proposition can be found in (Borodin and El-Yaniv, 1998, Theorem 4.4).

**Proposition 7.1** ((Fiat et al., 1991)). *For any randomized algorithm* ALG *for caching there is an input instance on universe of $k + 1$ pages such that the expected cost of* ALG *is more than $\ln k$ times the cost of the offline optimal solution.*

For a given algorithm, it constructs an instance consisting of marking phases, each with a single clean page such that the optimal algorithm pays 1 and the online algorithm pays at least $\ln k$.

Imagine an algorithm receiving at most $0.5 \, \text{OPT}$ predictions during this instance. Then, there must be at least $0.5 \, \text{OPT}$ phases during which the algorithm receives no prediction. Its cost is at least $\ln k$ in each such phase, giving total cost $0.5 \, \text{OPT} \ln k$.

Theorem 1.2 is implied by the following more general statement with $c = 1$ and $d = 0$.

**Theorem 7.2.** *Let $c \geq 1$ and $d \geq 0$ be constants. Any $(cf^{-1}(\eta) + d)$-smooth algorithm for caching with action predictions has to use at least $f(c^{-1} \ln k - d) \, \text{OPT}$ predictions.*

*Proof.* Consider a fixed algorithm accepting action predictions. Choose $T$ long enough, an arbitrary prediction for each time step $t = 1, \dots, T$, and give them to the algorithm at time 0. Having the predictions already, this algorithm becomes a standard randomized algorithm which does not use any further predictions. We use Proposition 7.1 to generate an instance such that $\mathbb{E}[\text{ALG}] \geq \text{OPT} \ln k$, where ALG denotes the cost of the algorithm with predictions generated in advance. It is clear that these predictions, generated before the adversary has chosen the input instance, are useless, not helping the algorithm to surpass the worst-case bounds. However, since the universe of pages has size only $k + 1$, each of the predictions can differ from an optimal algorithm by at most one page.

If we want to have $\frac{\mathbb{E}[\text{ALG}]}{\text{OPT}} \leq cf^{-1}(\frac{\eta}{\text{OPT}}) + d$, then we need

$$\frac{\eta}{\text{OPT}} \geq f\left(\frac{\mathbb{E}[\text{ALG}]}{c \, \text{OPT}} - d\right) > f\left(\frac{\ln k}{c} - d\right).$$

Since every prediction has error at most 1, we need to receive at least $\eta \geq f(c^{-1} \ln k - d) \, \text{OPT}$ predictions. □

## 7.2 MTS

Antoniadis et al. (2023) showed the following lower bound on smoothness of algorithms for general MTS with action predictions.

**Proposition 7.3** (Antoniadis et al. (2023)). *For $\eta \geq 0$ and $n \in \mathbb{N}$, every deterministic (or randomized) online algorithm for MTS on the $n$-point uniform metric with access to an action prediction oracle with error at least $\eta$ with respect to some optimal offline algorithm has competitive ratio $\Omega\left(\min\left\{\alpha_n, 1 + \frac{\eta}{\text{OPT}}\right\}\right)$, where $\alpha_n = \Theta(n)$ (or $\alpha_n = \Theta(\log n)$) is the optimal competitive ratio of deterministic (or randomized) algorithms without prediction.*

We use this proposition to prove the following theorem from which Theorem 1.4 directly follows.

**Theorem 7.4.** *For $\eta \geq 0$ and $n \in \mathbb{N}$, every deterministic (or randomized) online algorithm for MTS on the $n$-point uniform metric with access to an action prediction oracle at most once in $a$ time steps with error at least $\eta$ with respect to some optimal offline algorithm has competitive ratio $\Omega\left(\min\left\{\alpha_n, 1 + \frac{a\eta}{\text{OPT}}\right\}\right)$, where $\alpha_n = \Theta(n)$ (or $\alpha_n = \Theta(\log n)$) is the optimal competitive ratio of deterministic (or randomized) algorithms without prediction.*

*Proof.* We extend the $(n - 1)$-point uniform metric from the proposition above by a single point $p_\infty$ whose cost will be $+\infty$ at each time step, ensuring the optimal algorithm will never be located there. Consider a fixed algorithm and a predictor producing at most one prediction in $a$ time steps with the total prediction error $\eta$. By issuing prediction $p_\infty$ in all missing time steps, we complete predictions for each time step with error at least $\eta' \geq a\eta$.

By proposition above, the algorithm with completed predictions has competitive ratio at least

$$\Omega\left(\min\left\{\alpha_{n-1}, 1 + \frac{\eta'}{\text{OPT}}\right\}\right) \geq \Omega\left(\min\left\{\alpha_n, 1 + \frac{a\eta}{\text{OPT}}\right\}\right),$$

since $\alpha_n$ and $\alpha_{n-1}$ differ by at most a constant factor. □

## 8 FITF ORACLE

In this section we work with a predictor which tells us which page in our current cache will be requested furthest in the future, we call it a FitF page. Note that this is not the same as the predictions considered in Section 3, where we receive a page not present in Belady's cache. Belady evicts a FitF page from its current cache content which may be different from the FitF page from the current cache content of our algorithm. Prediction error is the total number of times the predictor reports an incorrect FitF page.

We split our algorithm into Follower and Robust part. The Follower (Algorithm 5), checks at each page fault whether Belady starting at the same time with the same cache content also has a page fault. If yes, it evicts a page reported by the predictor. Otherwise, it switches to the Robust part (Algorithm 6).

---

**Algorithm 5:** Follower with FitF oracle

1   $P :=$ starting cache content;
2   **foreach** *pagefault* **do**
3      Compute Belady for the sequence from the beginning of this execution starting with $P$;
4      **if** Belady *has page fault as well* **then**
5          $p :=$ page in the current cache chosen by the predictor;
6          evict $p$;
7      **else**
8          Run one phase of Algorithm 6 starting with the current cache content;

---

**Lemma 8.1.** *Consider one execution of Algorithm 5, denoting $\sigma$ the request subsequence and $\varphi$ the number of incorrect predictions received during this execution. Algorithm 5 pays the same cost as* Belady *serving $\sigma$ and starting with cache content $P$. There is a tie-breaking rule for* Belady *such that the cache contents of both algorithms after processing $\sigma$ differ in at most $\varphi$ pages.*

*Proof.* Whenever the algorithm has a page fault and Belady not, the execution of Algorithm 5 terminates. Therefore, both algorithms have the same cost during the execution.

Denote $A$ and $B$ the cache contents of our algorithm and Belady respectively. We choose the following tie-breaking rule for Belady: whenever the algorithm evicts $p \in A \cap B$ which is no more requested in $\sigma$, Belady evicts $p$ as well. The size of $A \setminus B$ increases only when the algorithm evicts a predicted page $p \in A \cap B$ and Belady evicts a different page $q \in A \cap B$. This can happen only if the next request of $p$ comes earlier than $q$ by the tie-breaking rule above. Since $p, q \in A$, the oracle made a prediction error. □

Robust part (Algorithm 6) uses a parameter $b$ which controls the number of predictions used during its execution. It runs for a duration of a single marking phase split into $\log k$ windows, as in Section 3, making sure that the number of predictions received in each window is at most the number of clean pages received so far. Evictions of random unmarked pages are used at page faults with no available prediction. At the end, it loads all marked pages. This is to ensure that the difference between the optimal and algorithm's cache content can be bounded by the cost of the optimal algorithm during the phase (using Observation 2.2) instead of accumulating over repeated executions of Follower and Robust.

**Lemma 8.2.** *Consider one execution of Algorithm 6 during which it receives $\varphi$ incorrect predictions. The expected cost incurred by Algorithm 6 is at most $2\Delta^B + 3\varphi(1 + b^{-1}\log k)$, where $\Delta^B$ denotes the cost incurred by* Belady *starting at the same time with the same cache content.*

---

**Algorithm 6:** Robust with FitF oracle

---

1   $P :=$ starting cache content;
2   $S := [t = k - 2^j + 1 \mid \text{for } j = \log k, \dots, 0]$;
3   $W_i := [S[i], S[i+1] - 1]$ for $i = 1, \dots, \log k + 1$ ;      // Split the phase into windows
4   **foreach** *pagefault at time t during the phase* **do**
5      $c_t :=$ number of clean pages which arrived so far;
6      **if** *number of received predictions in the phase is less than $bc_t$* **then**
7          **if** *number of received predictions in this window is less than $c_t$* **then**
8              $p :=$ page in the current cache chosen by the predictor;
9              evict $p$;
10     **else**
11        evict a random unmarked page;

12   Once phase has ended, load all marked pages to the cache and run Algorithm 5;

---

*Proof.* There are three kind of page faults:

1. evicted page is chosen by the predictor

2. requested page was chosen before by the predictor, evicted page was chosen at random

3. both evicted and requested pages were chosen at random

In the worst case, we can assume that once we run out of budget for predictions, all incorrectly evicted pages are requested in page faults of type 2 and returned to the cache. Now, let $g$ denote the number of pages evicted due to correct predictions – they are not going to be requested in this phase anymore (Observation 2.3). All other evicted pages are chosen uniformly at random among unmarked pages which were not evicted due to correct predictions. So, until another batch of page faults of type 1, we have only page faults on arrivals and the probability of a page fault on arrival $a$ is at most

$$\frac{c_a - g}{k - (a - c_t) - g},$$

where $c_a$ is the number of clean pages until arrival $a$ and $k - (a - c_t)$ is the number of unmarked pages, at most $g$ of them were evicted due to correct predictions.

We count the number of page faults in window $i$ for $i = 1, \dots, \log k + 1$. We denote $m_i$ the number of page faults of type 1 and resulting into eviction of $g_i$ correctly predicted pages. Then, by our assumption, we have $m_i - g_i$ page faults of type 2. The expected number of page faults of type 3 depends on when do types 1 and 2 happen. In the worst case, they all happen in the beginning of $W_i$ as well as all arrivals of clean pages. We consider three cases.

**Case A.**   Prediction budget was not depleted, there were only evictions of type 1.

$$\Delta^A(W_i) = m_i = \varphi_i + g_i.$$

**Case B.**   There were $m_i = c_{i+1}$ predictions during $W_i$ and we have $\varphi_i = c_{i+1} - g_i$. After page faults of type 2, there are at most $c_{i+1} - g_i$ randomly chosen unmarked pages evicted. Therefore, the expected number of page faults of type 3 is at most

$$\sum_{a \in W_i} \frac{c_{i+1} - g_a}{k - (a - c_{i+1}) - g_a} \leq \sum_{a \in W_i} \frac{c_{i+1} - g_i}{k - (a - c_{i+1}) - g_i}$$

$$\leq \sum_{a \in W_i} \frac{c_{i+1} - g_i}{k - a} \leq \frac{k}{2^i} \cdot \frac{c_{i+1} - g_i}{k/2^i} = c_{i+1} - g_i.$$

Therefore, counting evictions of types 1, 2, and 3, we have

$$\Delta^A(W_i) \leq (\varphi_i + g_i) + \varphi_i + (c_{i+1} - g_i) \leq g_i + 3\varphi_i.$$

**Case C.** There were $bc_{i+1}$ predictions since the beginning of the phase. We have $m_i \le c_{i+1}$ and $c_{i+1} - g_i \le \frac{1}{b}(bc_{i+1} - g_i) \le \frac{1}{b}\varphi$ where $\varphi$ is the total number of incorrect predictions received since the beginning of the phase. We have

$$\Delta^A(W_i) \le c_{i+1} + (c_{i+1} - g_i) + (c_{i+1} - g_i) \le g_i + 3(c_{i+1} - g_i),$$

which is at most $g_i + 3\varphi/b$.

Now, the sum of costs over all the windows is at most

$$\sum_i g_i + \sum_i 3\varphi_i + \sum_i 3\varphi/b + c \le 2c + 3\varphi + \frac{3\varphi}{b}\log k,$$

where $c = \sum_i c_i \le \Delta^B$, because we consider Belady starting with the same cache content as the algorithm which does not contain the clean pages. $\qquad\square$

**Theorem 8.3.** *Let $b \in \{1, \ldots, \log k\}$ be a parameter. During a request sequence with optimum cost* OPT, *our algorithm receives at most $O(b)$* OPT *predictions and its expected cost is always bounded by $O(\log k)$* OPT. *If only $\varphi$ predictions are incorrect, its expected cost is at most*

$$\left(2 + \frac{\varphi}{\text{OPT}}(4 + 3b^{-1}\log k)\right)\text{OPT}.$$

*Moreover, if $\varphi = 0$, its cost is equal to* OPT.

*Proof.* We split the time horizon into intervals corresponding to executions of Follower and Robust. For each interval $i$, we denote $\varphi_i$ the number of received incorrect predictions, $\Delta_i^B$ the cost incurred by Belady started with the same content as our algorithm and $\Delta_i^O$ the cost incurred by the optimal solution during interval $i$. We denote $F$ the set of intervals during which Follower was executed and $R$ the set of intervals during which Robust was executed. We also define $0 \in R$ an empty interval in the beginning of the request sequence with $\Delta_0^O = \Delta_0^B = 0$.

In order to prove bounds on robustness and number of used predictions, we provide relations between $\Delta_i^B$ and $\Delta_i^O$ independent of $\varphi$. For each $i \in F$, we have $i - 1 \in R$. Interval $i - 1$ is a marking phase and Robust has all marked pages in the cache at the end (Lemma 8.2). By Observation 2.2, the starting cache content of Follower in interval $i$ differs from optimal cache in at most $\Delta_{i-1}^O$ pages. Therefore, we have

$$\Delta_i^B \le \Delta_i^O + \Delta_{i-1}^O \quad \forall i \in F. \tag{12}$$

For each $i \in R$, we have $i - 1 \in F$ and $i - 2 \in R$. By Observation 2.4, the difference between the cache of Follower and optimum increases during interval $i - 1$ by at most $\Delta_{i-1}^O$. Since the starting cache of Follower in interval $i - 1$ differs from optimal in $\Delta_{i-2}^O$ pages, the starting cache of Robust in interval $i$ differs from optimum by at most $\Delta_{i-2}^O + \Delta_{i-1}^O$. Therefore, we have

$$\Delta_i^B \le \Delta_i^O + \Delta_{i-1}^O + \Delta_{i-2}^O \quad \forall i \in R. \tag{13}$$

Using equations equation 12 and equation 13, we can bound the number of used predictions as

$$\sum_{i \in F} \Delta_i^B + \sum_{i \in R} b\Delta_i^B \le 3b\,\text{OPT}.$$

Since $\varphi \le \Delta_i^B$, for $i \in F$, and $\varphi_i \le b\Delta_i^B$ for $i \in R$, we have the following robustness bound:

$$\begin{aligned}
\text{ALG} &\le \sum_{i \in F} \Delta_i^B + \sum_{i \in R}\left(\Delta_i^B + \varphi_i(3 + 3b^{-1}\log k)\right) \\
&\le \sum_{i \in F} \Delta_i^B + \sum_{i \in R} \Delta_i^B(1 + b)(3 + 3b^{-1}\log k) \\
&\le \text{OPT}\cdot O(\log k),
\end{aligned}$$

where the last inequality follows from equation 12, equation 13, and $b \le \log k$.

Now, we analyze smoothness. We can bound $\Delta_i^B - \Delta_i^O$ by the difference between optimal and algorithm's cache in the beginning of the interval $i$. This is at most $\varphi_{i-1}$ for each $i \in R$ (Lemma 8.1) and at most $\Delta_{i-1}^O$ for each $i \in F$ by equation 12. Lemmas 8.1 and 8.2 imply

$$
\begin{aligned}
\text{ALG} &\leq \sum_{i \in F} \Delta_i^B + \sum_{i \in R} \left( 2\Delta_i^B + \varphi_i(3 + 3b^{-1} \log k) \right) \\
&\leq \sum_{i \in F} (\Delta_i^O + \Delta_{i-1}^O) + \sum_{i \in R} \left( \Delta_i^O + \varphi_{i-1} + \varphi_i(3 + 3b^{-1} \log k) \right) \\
&\leq 2\,\text{OPT} + \varphi(4 + 3b^{-1} \log k).
\end{aligned}
$$

1-consistency of our algorithm can be seen from the fact that each execution of Robust is triggered by an incorrect prediction. Therefore, with perfect predictions, only Follower is used and behaves the same as Belady. $\qquad\square$

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

## A  COMPUTATIONS FOR TABLE 1

In this section, we present the computations for the numbers of predictions we obtained in Table 1.

For $f(i) = i$ and $f(i) = i^2$, we have $f(\log k)$ equal to $\log k$ and $\log^2 k$ respectively.

For $f(i) = 2^i - 1$, we identify the first window $i$ longer than $f(i) - f(i-1)$. Note that the length of window $i$ is $k/2^i = 2^{\log k - i}$ and this is equal to the sum of lengths of the windows $j > i$. The total number of predictions used will be therefore $f(i) + 2^{\log k - i}$. For $i = \log \sqrt{k} + 1$, we have $f(i) - f(i-1) = 2^i = 2^{\frac{1}{2}\log k + 1} > 2^{\log k - i}$. Therefore, we use $2^i - 1 + 2^{\log k - i} \geq 3\sqrt{k}$ predictions in each robust phase. Since offline optimum has to pay at least $1$ per robust phase, we use at most $O(\sqrt{k})$ OPT predictions in total.

For $f(i) = 0$, we ask for a prediction at each arrival of a clean page. The number of queries used will therefore be at most the number of clean arrivals, which is at most $2\,\text{OPT}$.

