# OpenReview forum: "Algorithms for Caching and MTS with reduced number of predictions"
_ICLR.cc/2024/Conference — ICLR 2024 poster_

### Official Review · Reviewer_utqC · 2023-10-27

**Soundness:** 3 good
**Presentation:** 3 good
**Contribution:** 3 good
**Rating:** 8
**Confidence:** 3

**Summary:**

This paper considers online caching and metrical task systems (MTS) in the popular algorithms with predictions framework.  Following recent related work due to Im et al. 2022, they consider reducing the number of predictions utilized by the algorithm.  This paper differs from that one in two respects:
 - This paper considers action predictions as introduced by Antoniadis et al. which are different from the usual notion of "next-arrival-time" predictions as used by Lykouris and Vassilvitskii 2021, Rohatgi 2020, and Im et al. 2022.
- In addition to considering bounded number of predictions as in Im et al. 2022, they also propose studying well-separated queries to the predictions.  In this setting there is a lower bound on the number of time steps between queries to the predictions.  This is motivated by settings where we may not be able to immediately query the prediction.

The main theoretical results are as follows:
- An algorithm for caching with robustness $O(\log k)$, smoothness $O(f^{-1}(\eta / OPT))$ and using at most $O(log k)OPT$ queries to predictions.  $f$ is an increasing function with some restrictions.  Examples of the relationship for different $f$'s are described.
- A lower bound of $f(\ln k) OPT$ queries to the predictor for any algorithm that is $f(\eta)$ smooth.
- For MTS, the authors study well-separated queries that arrive once every $a$ steps and give an algorithm with cost $O(a)(OPT + 2\eta)$ and give a nearly tight lower bound.

To complement these, several experiments using datasets that have been utilized in prior work are carried out, comparing the proposed algorithm to the algorithms proposed in several prior works.

**Strengths:**

- Developing algorithms which make use of predictions economically is an interesting question that has arisen recently
- The result for caching gives a nice way of quantifying the trade-off between number of predictions and smoothness through the function $f$, although it is a little more difficult to interpret than say the result in Im et al. 2022.
- The experimental results are promising

**Weaknesses:**

- The writing and grammar needs improvement at certain points (see below for more specific comments).

**Questions:**

### Major Questions and Comments

- The motivation for well-separated queries is settings in which the waiting for the query to be produced may take a number of time steps to produce.  Have you considered a setting where the prediction queries are explicitly delayed?  For example, based on the state of the algorithm at time $t$, we make a query to the predictor, but the result does not arrive until time $t+a$.  So predicted information is directly more costly in the sense that it may no longer be "fresh" or as useful when we receive it.
- For MTS, you are unable to give a bound on the number of predictions that scales with $OPT$ since the adversary can scale the instance to make $OPT$ arbitrarily small.  Can the absolute number of predictions be bounded?  For example is it possible to use $o(T)$ queries to the predictions for this setting, or is this impossible similar to the lower bound in Theorem 1.4?


### Minor Comments
 - First page, third paragraph - "Online nature of ..." -> "The online nature of ..."
 - Second page, second new paragraph - "Using method of Blum and Burch ..." -> "Using the method of Blum and Burch..."
 - Several places - In case of caching ..." -> "In the case of caching"
 - Please make general improvements to the grammar.
 - Page 6, first paragraph - "$f(i) \leq 2^j - 1$" - should this be $f(i) \leq 2^i - 1$?  I don't see $j$ defined nearby...
 - For the synthetic noise a log-normal distribution is used.  I believe you meant the parameters for this distribution are $\mu = 0$ and $\sigma \in [0,50]$.  The mean of a log-normal distribution cannot be 0...
 - In the captions for Figures 1 and 2, please clarify if the standard deviation reported is for the competitive ratio averaged over many trials (and not the noise for the synthetic predictions), and how many trials were used for this average if so.

---

> ### Author Response · Authors · 2023-11-22
>
> Thank you for your questions and comments. We have fixed the grammar and some
> typos which we have encountered when revising our manuscript.
>
> Major Questions and Comments:
>
> * We have not considered the case where the predictions are explicitly delayed,
>   but we find it a very interesting question. This setting brings additional
>   challenge to our analysis, where the prediction error should be computed
>   with respect to the state of Belady at the time step when the predictor
>   was queried rather than the time step when the prediction was produced
>   (and used).
>
> * It is not possible to achieve non-trivial results for MTS with o(T) predictions.
>   To illustrate this, imagine an input sequence composed of a large number $N$
>   of difficult MTS instances of constant length one after another.
>   Each of the difficult MTS instances requires a certain number of predictions
>   and the total number of required predictions then needs to scale linearly with
>   $N$. An additional note: although OPT might be much smaller than $T$ in some
>   cases, it may also grow linearly in $T$.
>
> Minor Comments:
>
> * On page 6, first paragraph, there should be $f(i)\leq 2^i-1$ as you suggest.
>
> * $\mu, \sigma$ are the parameters of the normal distribution. We have fixed our explanation in the manuscript.
>
> * The plots and tables in Section 5 contain averages of 10 independent
>   experiments. Unfortunately, this line slipped out of the main version and was
>   included only in Appendix. The standard deviation refers to the competitive
>   ratio and the number of requested predictions respectively.
>   We have added the omitted explanation.

---

> > ### Comment · Reviewer_utqC · 2023-11-22
> >
> > Thank you for the response.

---

### Official Review · Reviewer_bwNg · 2023-10-28

**Soundness:** 4 excellent
**Presentation:** 4 excellent
**Contribution:** 4 excellent
**Rating:** 8
**Confidence:** 4

**Summary:**

This paper studies online algorithms for caching (and more generally MTS) with predictions. The considered prediction is “action prediction”, which tells what (the ideally optimal) algorithm does. The goal here is not only to achieve a good prediction error vs competitive ratio tradeoff, but also to use fewer predictions, through queries to the action prediction.

The results look very general and strong. The main results are about caching with size k. It shows for a large class of functions f, the algorithm uses f(log k) * OPT predictions, achieves consistency 1, robustness O(log k) and smoothness O(f^{-1}(\eta / OPT)), where \eta is the number of wrongly predicted actions. Furthermore, these bounds are also justified by nearly matching lower bounds.

Technically, the caching algorithm first tries to follow the prediction, and if the prediction “looks bad”, it switches to a modified Marking algorithm to preserve the robustness. But to preserve consistency, this modified Marking is run for only O(k) distinct requests and will switch back to follow the prediction again. This algorithm is intuitive, but the analysis is very involved, especially to deal with the randomness of the Marking algorithm and to control the number of predictions used.

For the MTS problem, it does not make much sense to aim for a query complexity bound like f(\cdot) OPT, and the paper turns to put a constraint that the algorithm can only make a query every $a$ steps, where $a$ is some parameter. This also makes the techniques very different from the caching algorithm.

**Strengths:**

- I’m impressed by the results and techniques.

- The paper is very clearly written, and the comparison to previous works is adequate.

- This paper has a very comprehensive discussion of how to potentially implement the predictor and the ways to tune the algorithm for better practical performance — I find this very helpful.

**Weaknesses:**

- The MTS part does not seem to connect well enough to the rest of the paper which mainly focuses on caching; especially I cannot see the tight relation in terms of algorithms/techniques.

- The experiments only show a marginal advantage over baselines on real datasets.

**Questions:**

- In Section 1.1, it is not clear enough what \eta means without reading later parts of the paper, particularly what it means by “total error” (mentioned in the first paragraph of Section 1.1). Some intuitive explanations would be helpful. Furthermore, it may look like the meaning of \eta is different between Theorem 1.1 and Theorem 1.3, and this should be clarified, too. (By the way, this is not clearly discussed even in Section 2.1.)

- Theorem 1.2: \eta is not quantified; I guess we should say “for every \eta > 0”?

- In Section 2.2, it is assumed that the predictor is trying to simulate Belady. Is this without loss of generality?

- In Algorithm 2, is $S$ a set or a sequence? I don’t find S = [ xxx ] a standard notation. Also, S[I] does not look standard, either.

- In Algorithm 2, line 4, is this to define $F$?

- In Algorithm 2, line 4, how is p chosen?

- In Algorithm 2, I find additional whitespace before “;” in lines 6 - 8.

- Section 3, it’s better to give some “comment” in the algorithm (or somewhere in the main text) to describe what F and S stand for.

- Page 6, there’s a paragraph about “implementation suggestions”. This paragraph is not directly related to the proof of the main theorem, so it is slightly out of context. Maybe move it to the end of the section, or somewhere in the experiments section?

- Page 7, 2nd paragraph. What is G_{i, I+1} exactly? In particular, what does “gap” mean? I’m not even sure if this is a number or a subset when I first read it.

- Page 7, 2nd paragraph. What do you mean by “period” X? Maybe you simply say X is a sequence of requests.

- I tried to read the proof of Lemma 3.3, but I didn’t find a clear place for “Robust_f uses at most f(log k) predictions”. Where is it? Also, does this bound depend on the randomness?


- Section 5, for the synthetic predictions, why do we consider the log-normal distribution?

---

> ### Author Response · Authors · 2023-11-22
>
> Thank you for your comments and questions.
> We have used them to improve our presentation and addressed them in our
> manuscript by including additional clarifications.
> Here, we state the clarifications explicitly.
>
>
> * The total prediction error $\eta$ is defined as the sum of errors of
>   all predictions. Formally, $\eta = \sum_{t=0}^T d(p_t, o_t)$. Here, $T$ is the
>   length of the input sequence and $p_t , o_t$ denote the state of our predictor and
>   the state of $OFF$ respectively at time $t$. Theorem 1.1 requires $OFF$ to be the
>   Belady's algorithm. The bound in Theorem 1.3 does not have this restriction
>   and holds for an arbitrary offline algorithm $OFF$.
>
> * We say that an algorithm is $f^{-1}(\eta)$-smooth if, for every $\eta>0$,
> its competitive ratio with predictions of error $\eta$ is bounded by $f^{-1}(\eta)$.
> We show that any such algorithm needs at least $f(\ln k) OPT$ predictions.
> Our lower bound is based on a situation when most of the predictions
> are incorrect.
>
> * We assume that the predictor is trying to simulate Belady because the
>   prediction error is measured with respect to Belady. Note that the current
>   predictors do try to imitate Belady (Liu et al. 2020).
>   Said that, it would be nice if
>   we could extend our analysis to the case where we do not know
>   which offline algorithm is simulated by the predictor. This would result
>   in a stronger kind of guarantee as in Theorem 1.3.
>
> * Questions about Algorithm 2: $S$ is a set. We define $F$ in line 4.
>   $f$ in line 4 is a parameter of the algorithm. In case you meant
>   to ask about $p$ in line 4 of Algorithm 1, this $p$ is chosen arbitrarily
>   among pages in the algorithms cache not belonging to the predictor's cache
>   $P$.
>
> * We believe that the paragraph on implementation suggestions is related to the
>   algorithm's description and we prefer to leave it in its current place.
>
> * gap $G_{i-1,i}$ denotes a time interval between two executions of Robust.
>   Similarly, we meant $X$ to be a time period.
>
> * Algorithm Robust queries the predictor only on arrivals belonging to $F$
>   (Algorithm 2, line 6) and the size of $F$ does not depend on randomness
>   in the algorithm. When considering your question, we have noticed that we
>   have missed a single query in the last window $W_{\log k +1}$ in our
>   computation. So, $F$ is chosen in line 4 of Algorithm 2 to have size
>   at most $f(\log k)+1$. Therefore, each execution of Algorithm 2
>   requires at most $f(\log k)+1$ predictions.
>
> * We have chosen log-normal distribution in order to produce empirical results
>   readily comparable to those from previous works.
>   Noise from the log-normal distribution simulates rare but large failures of
>   the learning algorithm which we might expect from real-world predictors.

---

### Official Review · Reviewer_NjU5 · 2023-10-29

**Soundness:** 3 good
**Presentation:** 3 good
**Contribution:** 4 excellent
**Rating:** 8
**Confidence:** 4

**Summary:**

The paper studies the online caching problem and the metrical task system problem which admits online caching as a special case. In the online caching problem, we are given a cache of fixed capacity $k$ and a sequence of online arriving elements, and want to design an online algorithm that decides which elements to keep in the cache and which ones to evict when the cache becomes full. The goal is to minimize the number of cache misses, which occur when an arriving element is not present in the cache. In the metrical task system problem, we are given a metric space $M$ of states and an initial state $x_0\in M$. At each time step $t$, we receive a cost function $\ell_t: M \rightarrow R^+ \cup \set{0, \infty }$ and need to decide which state to move to. When moving to a state $x_t\in M$, we have to pay $\ell_t(x_t) + d(x_{t-1},x_t)$. The goal is to minimize the total payment. To see that online caching is its special case, we can view the elements kept in the cache as the state in a metric space.

The two problems are considered in the learning-augmented setting, where we can access imperfect predictions about the future.  More precisely, the authors assume that at any time $t$, if they query a predictor, the predictor returns a prediction $\hat{s}_t$ of the state that the optimal solution (or a good approximation solution) moves to. Since the prediction is imperfect, they define the prediction error $\eta$ to be the moving distance between the predicted state and the real optimal state. The authors show that for online caching,  there exists an algorithm that achieves $1$-consistency, $O(\log k)$-robustness, and $O(f^{-1}(\eta /OPT))$-smoothness within a query complexity at most $f(\log k)OPT$, where function $f( \cdot )$ is an increasing convex function with $f(0)=0$ and $f(i)\leq 2^i-1$ $\forall i\geq 0$. For metrical task system, the authors show that there exists an algorithm that achieves a cost at most $O(a) \cdot (OPT+2\eta)$ by making a query each $a$ time step. The smoothness lower bounds are further provided in the paper to show the optimality of their results. Finally, the proposed algorithms are evaluated empirically.

**Strengths:**

This is a technically solid paper. Online caching and MTS are very classical and important problems in online optimization, and can capture many real-world applications. The paper advances in the direction of augmenting online algorithms by restricted learning. The two problems are good targets since the existence of a robust framework for them alleviates the concern about robustness. The authors show non-trivial trade-offs between the query complexity and the smoothness, which may influence future work in the field of learning-augmented algorithms.

**Weaknesses:**

One weakness is that the action prediction setting still looks a little bit weird. To me, particularly in online caching, predicting the next arrival time of the current element makes more sense and is more learnable. It seems hard to learn the optimal actions from historical data in practice. But I understand that this is a setting proposed in the previous work, and the local information predictions (e.g. next arrival time) may not help in the restricted-learning setting.

**Questions:**

(1) The key algorithmic idea for algorithm 1 is built on observation 2.1 that Belady is somehow stable. If the predictor simulates other unstable algorithms, is there any non-trivial smoothness bound?

---

> ### Author Response · Authors · 2023-11-22
>
> We thank you for you comments.
>
> * Our smoothness bound for caching depends on the prediction error with respect
>   to Belady. This is not the case in our result for general MTS, where the
>   prediction error can be measured with respect to an arbitrary offline algorithm.
>   This result covers caching as a special case of MTS, but gives only a linear
>   smoothness.
>   We believe that accommodating non-Belady predictors in our result for caching
>   is a matter of a more refined analysis. Currently, we use properties specific
>   to Belady (Observations 2.1 and 2.3) to detect incorrect predictions in the
>   Follower part of our algorithm and estimate the prediction error in the anlysis
>   of the Robust part. Overcoming these issues seems interesting but challenging.
>
> * We do not want to argue that one type of predictions is better than another,
>   but we would like to point out several nice properties of action predictions.
>   First, they can be produced by any method which is able
>   to generate next-arrival predictions. This is due to the simple conversion
>   rule proposed by Antoniadis et al. (2020). Note that this does not
>   work the other way. For example,
>   a classification problem of determining whether a page is cache-friendly or
>   cache-averse as in predictors by Jain and Lin (2016) and Shi et al. (2019)
>   gives rise to action predictions but its solution does not contain enough
>   information to reconstruct next-arrival time predictions.
>   Second, action predictions are usable beyond caching. Note that the effect of
>   next-arrival time predictions is limited already for weighted caching,
>   where the best competitive ratio achievable with perfect next-arrival time
>   predictions is far from 1 and depends on the number of weight classes in the
>   input instance, see Bansal et al. (2022).
>   We believe that our paper adds a third item to this list, namely better
>   consistency bounds achievable in a regime with a limited access to the predictor.

---

> > ### Comment · Reviewer_NjU5 · 2023-11-22
> >
> > The reviewer thanks the authors for their response.

---

### Official Review · Reviewer_dLjd · 2023-10-29

**Soundness:** 3 good
**Presentation:** 3 good
**Contribution:** 3 good
**Rating:** 8
**Confidence:** 3

**Summary:**

This paper considers the problem of Metrical Task Systems (MTS) under the prediction setting. In the classical MTS problem, the algorithm starts from an initial state (or a point in a metrical space), and the adversary releases a cost function at each time step. After seeing the cost function, the algorithm needs to make a decision at each time: either stay in the same state or move to another state, which is possibly cheap, but some extra moving costs will be charged. In the prediction setting, the algorithm accesses a prediction on future information. In this paper, the prediction is considered the next stage of an optimal algorithm. Unlike other prediction papers, this work investigates how the number of predictions affects the algorithm.

The main contribution of this work is a learning-augmented framework achieves a good trade-off between the smoothness ratio and the number of predictions. For the Caching problem, this framework achieves a nearly optimal tradeoff. The authors also show that such a framework works for the general MTS problem. Besides this, this work also extends the framework for the setting where the prediction can also be obtained by at least a time step. Finally, the authors verify their algorithm in some datasets.

**Strengths:**

Strengths:

1. The paper is well-written and well-structured. I appreciate that the authors provide sufficient intuitions on appropriate places, which significantly improves the readability of the paper.
2. I particularly like the model where the prediction cannot be obtained by at least some fixed time step. This idea seems new and can capture many applications in practice.
3. The technical contribution is solid. Although the high-level picture of the proposed algorithm follows a typical framework in a learning-augmented area, the algorithm contains sufficient new ideas and analysis.

**Weaknesses:**

Weaknesses:

I don't see any major weaknesses in the paper. My main concern is that due to its technical nature, one would really be required to check all proofs for correctness. I am unfortunately not able to do that due to the time limit for reviewing. I only checked a few lemmas and I believe they are correct. If all proofs are correct, then I think this is a good paper.

**Questions:**

I don't have any specific questions.

---

> ### Author Response · Authors · 2023-11-22
>
> We thank you for your review and for your comments.

---

### Meta-Review · Area_Chair_oVRY · 2023-12-07

**Metareview:**

The paper studies the caching and metrical task system problems in the setting where the algorithm is given access to predictions. Similarly to previous work, the paper considers an action prediction model, and the main focus is on reducing the number of predictions used by the algorithm. The problems studied are classical online optimization problems and the specific setting considered in this work is well-motivated.  The reviewers appreciated the contributions of this work, and there was clear consensus that this paper makes a strong theoretical contribution to this line of work.

**Justification For Why Not Higher Score:**

The contribution may not be of broad enough interest to be highlighted as a spotlight.

**Justification For Why Not Lower Score:**

The theoretical contributions of the paper are strong and it is a valuable addition to the design and analysis of online algorithms that incorporate predictions.

---

### Decision · Program_Chairs · 2024-01-16

Accept (poster)